# Approach to map nanotopography of cell surface receptors

Christian Franke [1,2,3,9,10 ✉], Tomáš Chum[4,9], Zuzana Kvíčalová[4], Daniela Glatzová[4,5], Gregor Jörg Gentsch [1], Alvaro Rodriguez [6], Dominic A. Helmerich[7], Lucas Herdly[8], Harsha Mavila[4], Otakar Frank [6], Tomáš Brdička[5], Sebastian van de Linde [8,10 ✉] & Marek Cebecauer [4,10 ✉]

Cells communicate with their environment via surface receptors, but nanoscopic receptor organization with respect to complex cell surface morphology remains unclear. This is mainly due to a lack of accessible, robust and high-resolution methods. Here, we present an approach for mapping the topography of receptors at the cell surface with nanometer precision. The method involves coating glass coverslips with glycine, which preserves the fine membrane morphology while allowing immobilized cells to be positioned close to the optical surface. We developed an advanced and simplified algorithm for the analysis of single-molecule localization data acquired in a biplane detection scheme. These advancements enable direct and quantitative mapping of protein distribution on ruffled plasma membranes with near isotropic 3D nanometer resolution. As demonstrated successfully for CD4 and CD45 receptors, the described workflow is a straightforward quantitative technique to study molecules and their interactions at the complex surface nanomorphology of differentiated metazoan cells.

[1] Institute of Applied Optics and Biophysics, Friedrich Schiller University Jena, Jena, Germany. [2] Jena Center for Soft Matter, Friedrich Schiller University Jena, Jena, Germany. [3] Abbe Center of Photonics, Friedrich Schiller University Jena, Jena, Germany. [4] Department of Biophysical Chemistry, J. Heyrovsky Institute of Physical Chemistry, Czech Academy of Sciences, Prague, Czech Republic. [5] Laboratory of Leukocyte Signalling, Institute of Molecular Genetics, Czech Academy of Sciences, Prague, Czech Republic. [6] Department of Electrochemical Materials, J. Heyrovsky Institute of Physical Chemistry, Czech Academy of Sciences, Prague, Czech Republic. [7] Department of Biotechnology and Biophysics, Biocenter, University of Würzburg, Würzburg, Germany. [8] Department of Physics, SUPA, University of Strathclyde, Glasgow, UK. [9] These authors contributed equally: Christian Franke, Tomáš Chum. [10] These authors jointly supervised this work: Christian Franke, Sebastian van de Linde, Marek Cebecauer. ✉email: christian.franke@uni-jena.de; s.vandelinde@strath.ac.uk; marek.cebecauer@jh-inst.cas.cz

Supramolecular complexes drive numerous vital processes in cells, such as gene expression, metabolism, or molecular transport. Cellular membranes provide an excellent platform to assemble molecules into complex structures. Indeed, membrane-associated molecules, including surface receptors, have been found to form clusters with nanometric dimensions[1–5]. Lateral interactions and/or actin-cytoskeleton anchorage drive clustering of some receptors (e.g., tetraspanin platforms, focal adhesion complexes). However, the mechanism of cluster assembly for other receptors remains unknown and proposed theories are often controversial (e.g., lipid rafts). Indeed, receptor clustering is the subject of intense debate[6–8]. Several works have provided evidence for the monomeric character of receptors, which were shown to cluster using different experimental setups[9–11]. Thus, understanding their origins can help to unravel the very existence of receptor clusters. In this work, we developed a new approach for monitoring receptor nanotopography on the surface of lymphocytes, but the methods and general principles discussed herein apply to the surface of any cell.

A common feature of supramolecular structures, including membrane receptor clusters, is their size (<200 nm), too small to be analyzed by standard light microscopy. Therefore, molecular assemblies in cells are often studied indirectly. For example, total internal reflection fluorescence (TIRF) microscopy enabled visualization of dynamic receptor microclusters in B and T cells[12,13]. It was the lateral mobility of these entities that indicated the very existence of clusters in these cells. The size and shape of the observed clusters were irresolvable because the spatial resolution of TIRF microscopy is diffraction limited. Later, super-resolution (SR) microscopy techniques were developed to surpass the diffraction limit and offered a more detailed insight into the architecture of cell receptor clusters (for example, refs. [5,14–16]).

Currently, single-molecule localization microscopy (SMLM) is the method of choice to study the organization of membrane receptors due to its ability to localize emitters (i.e., labeled receptors) with nanometer precision and its potential for quantitative assessments of protein distribution and number[17–20]. SMLM-based studies confirmed the earlier observations using TIRF microscopy that receptors and associated signaling molecules can cluster in non-stimulated and stimulated immune cells[21–25]. However, there is an intense discussion about the feasibility of SMLM for the cluster analysis of membrane molecules[26–28]. Procedures aiming to minimize the impact of methodological artifacts on the SMLM results were developed[29–31].

In most of these studies, the SMLM methods were used to generate localization maps by projecting the presumably three-dimensional receptor distribution onto a two-dimensional plane. The precise information about the axial position of emitters (e.g., receptors), and therefore their distribution, is missing. This was an accepted trade-off due to the limited axial resolution of the applied SR methods (including STED and SIM; ref. [32]). Moreover, the plasma membrane is routinely depicted as a rather featureless structure (for example, refs. [33,34]). Yet differentiated cells of vertebrates are densely covered with membrane protrusions and invaginations, thus resulting in a highly three-dimensional surface[35]. For example, scanning electron microscopy micrographs have demonstrated that finger-like membrane protrusions reminiscent of microvilli dominate the surface of lymphocytes[36–38]. T-cell microvilli are dynamic structures ~100 nm in diameter and 0.5–5 μm long[39,40]. Although less well understood on immune cells, tips of microvilli can potentially accumulate membrane receptors in domains with a diameter ~100 nm, analogous to signaling receptors and channels on epithelial and sensory cells[41].

Electron microscopy (EM), with its ability to provide information about ultrastructural details, was the key method for microvilli characterization in the pioneering works[42]. However, EM cannot visualize specific proteins efficiently, since the labeling densities are limited by ligand/antigen accessibility, steric hindrance, and electron repulsion[43–45]. A rapid development of SR light microscopy techniques enabled the visualization of three-dimensional objects with high precision, with the use of highly specific and frequently efficient labeling methods and on living cells (e.g., refs. [46,47]). However, to characterize receptor distribution on nanometric membrane structures such as microvilli, it is crucial to develop three-dimensional SR techniques with axial resolution well below 50 nm. Several SMLM-based methods have been suggested[48], such as astigmatism[49], multiplane[47] or biplane imaging[50,51], double-helix point spread function[52] or interferometric PALM[53].

Although microscopy techniques with dramatically improved spatial resolution recently became available, the (cell) sample preparation for SR imaging still suffers from several caveats. The spatial resolution offered by SMLM comes with the tradeoff of time resolution since thousands of camera frames are needed to render a map sufficiently representing, for example, a receptor distribution on a cell[54]. Thus, the movement of imaged objects (e.g., proteins) must be minimized. This is usually accomplished by the fixation of cells prior to imaging. Importantly, cells grown or isolated initially in suspension must be attached to the optical surface (e.g., coverslips). Single-molecule fluorescence microscopy techniques (including SMLM) depend on efficient transmission of photons from emitters and thus, provide the best results for molecules close to the optical surface, especially when aiming for a subsequent quantitative analysis[48]. Poly-L-lysine (PLL, or its isomer poly-D-lysine) is used for the immobilization of suspension cells in standard protocols[55]. However, attachment to a surface via interaction with PLL leads to the deformation of cells[56]. Similarly, the flattening of cell surfaces is observed when positioning immune cells on adhesion molecules (e.g., ICAMs) directly coated on coverslips or linked to a supported planar lipid bilayer[57,58]. Currently, none of the available methods can immobilize cells close to the optical surface without interfering with their complex surface morphology.

Here, we describe a new method for mapping nanotopography of receptors at the plasma membrane of T cells with near isotropic three-dimensional resolution. The workflow involves a newly developed, optimized coating of glass coverslips, which preserves the complex morphology of a cell surface while allowing for the positioning of immobilized cells close to the optical surface. Moreover, we advanced the recently reported TRABI method[59] for nanometer precise three-dimensional SMLM imaging to improve data acquisition, processing and drift correction, as well as the integrity of the three-dimensional SMLM data set. This allowed us to employ a straightforward quantitative assessment of the axial distributions of selected cell surface receptors at the nanoscale. We applied this method to reveal the three-dimensional nature of CD4 receptor clusters in non-stimulated Jurkat cells. We also demonstrate the importance of membrane protrusions for the segregation of CD45 phosphatase from CD4 involved in T-cell signaling. The presented method enables molecular studies of cell surface nanomorphology but also of other fluorescent nanoscopic structures.

## Results

**Preservation of membrane morphology during cell immobilization on coverslips**. To map receptor distribution at the complex surface of cells and the nanoscale, the experimental setup must include quantitative labeling of target molecules, an appropriate sample preparation, and a nanometer precise three-dimensional imaging method. In this work, labeling of surface

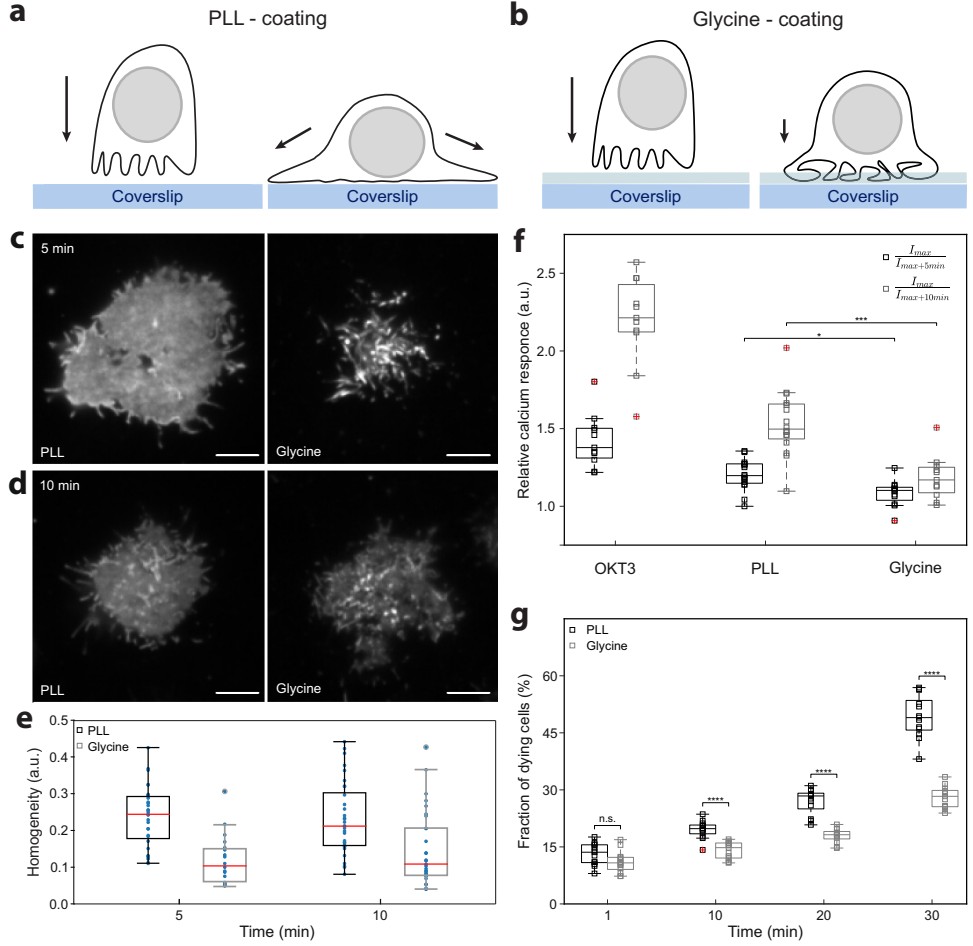

**Fig. 1 Coating of coverslips with glycine better preserves cell surface morphology and resting state of immobilized T cells. a**, **b** Schematic illustration of T-cell landing on PLL- (left panels) and glycine-coated coverslips (right panels). The arrows indicate forces influencing the cell on the coated coverslip. The g-force is the sole force affecting T cells on glycine. T cells on PLL are further stretched due to electrostatic interactions of the surface molecules with PLL. The blue stripes represent glass coverslip, the light blue stripes above represent the glycine layer (not to scale). **c**, **d** Live-cell TIRF microscopy of CD4-GFP in Jurkat cells landing on PLL- (left panels) or glycine-coated (right panels) coverslips measured at 37 °C. Representative contact morphology is shown for cells 5 min (**c**) and 10 min (**d**) after addition of the cell suspension to coverslips. **e** Graph showing signal homogeneity of cell contacts on PLL- and glycine-coated coverslips as in **c** and **d**. Signal homogeneity was quantified according to ref. [91] (see "Methods" and Supplementary Data for more details). In total, 31 cells for PLL$_{5min}$, 21 cells for Glycine$_{5min}$, 32 cells for PLL$_{10min}$ and 29 cells for Glycine$_{10min}$ were analyzed. Blue dots represent values from individual cells, red lines median values, boxes indicate upper and lower quartile of data and outliers are circled. Statistical significance was evaluated using one-way ANOVA test; $p_{5min} < 10^{-5}$, $p_{10min} \leq 0.0025$. **f** Calcium response induced by the interaction of Jurkat cells with coverslips coated with stimulating antibody (anti-CD3ε; OKT3), PLL or glycine as indicated by the changes in the fluorescence of ultrafast genetically encoded calcium sensor GCaMP6f$_u$ at 5 min (black bars; $I_{max}/I_{max+5\,min}$) and 10 min (gray bars; $I_{max}/I_{max+10\,min}$) after the maximal stimulation of cells ($I_{max}$). The extent of calcium response was calculated from the signal decay after cell spreading on stimulating surfaces (see "Methods" and Supplementary Data for more details). Relative calcium response equal 1 indicates no stimulation. Higher values indicate measured stimulation of cells. Small squares represent values calculated for individual cells; solid lines median values, boxes indicate upper and lower quartile of data and outliers are marked with a red cross. Statistical significance was evaluated using one-way ANOVA test; *$p < 0.05$ or ***$p < 0.001$. Five examples of full signal traces together with graphical representation of data processing are shown in Supplementary Fig. S7. **g** Viability of Jurkat cells interacting with coverslips coated with PLL- (black bars) or glycine-coated (gray bars) coverslips. Dying cells were defined as a fraction of 7-aminoactinomycin-positive cells within the imaged area using wide-field microscopy (see "Methods" and Supplementary Data for more details). Values at 1 min represent the starting point for the analysis—a minimal period required for cells to land at the optical surface. Small squares represent individual measurements, solid lines median values, boxes indicate upper and lower quartile of data and outliers are marked with a red cross. Statistical significance was evaluated using unpaired t-test with unequal variances (Satterthwaite's approximation); ****$p < 0.0001$, n.s. (not significant). Scale bars, 5 μm.

receptors was performed by standard immunofluorescence protocols using directly labeled, highly specific primary antibodies to avoid artificial clustering by high level crosslinking with secondary antibodies[60–62]. To prevent visualization of intracellular molecules, fixed Jurkat cells were labeled in the absence of membrane permeabilization.

Cells grown or isolated in suspension, e.g., lymphocytes, must be immobilized on the optical surface (e.g., coverslips; Fig. 1a, b)

to facilitate imaging approaches, which require long acquisition times. This is true for most SMLM techniques[63]. Commonly, PLL-coating of glass coverslips is used to adhere suspension cells to the optical surface[56]. Negatively charged biomolecules at the cell surface electrostatically interact with the polycationic layer formed by PLL on coverslips[64,65]. However, such interaction can deform the cell surface (Fig. 1a; ref. [56]). Indeed, we have repeatedly observed a rapid cell spreading and flattening of its

surface upon settling on the PLL-coated coverslip, visualized by live-cell TIRF microscopy of CD4-GFP fusion protein (Fig. 1c, d; Supplementary Fig. S1 and Supplementary Movie 1). CD4-GFP distributes evenly on Jurkat cell plasma membrane analyzed by confocal microscopy (Supplementary Fig. S2). The live-cell data indicate that, with small exceptions, a homogenous distribution of CD4-GFP signal dominated in such cells within 1–5 min after the first detectable contact with the optical surface (Supplementary Fig. S1). Continuous and homogenous fluorescence signal indicated a lack of CD4-GFP accumulation on membrane protrusions and invaginations. These data suggest that lymphocytes immobilized on PLL-coated coverslips exhibit prevalently flat surface morphology.

We thus required an alternative method for coating of coverslips, which better preserves their native morphology. Coating of an optical surface with glycine has been previously used to minimize non-specific background signals in single-molecule fluorescence imaging[66,67]. However, when applied without prior PLL treatment, glycine formed a gel-like layer on a glass surface (Supplementary Fig. S3). Our atomic force microscopy (AFM) measurements indicated a continuous surface coating with glycine (Supplementary Fig. S3b). However, the AFM measurements were limited by the softness of the glycine layer and a tendency of this material to adhere to the AFM tip. We were thus unable to determine the exact thickness and stiffness of the glycine layer. The few holes, probably caused by air bubbles, observed in the tested samples indicated that the average thickness of the glycine layer was at least 15 nm (Supplementary Fig. S3c). We further observed crystals on coverslips coated with glycine that were subsequently dried. On the contrary, no such precipitate was observed on dried PLL-coated coverslips (Supplementary Fig. S3d). We thus conclude that glycine forms a narrow, gel-like structure on the glass surface, which functions as a semi-soft cushion upon cell landing.

To investigate the impact of glycine coating on cell surface morphology, we imaged living Jurkat cells expressing CD4-GFP during landing on PLL- or glycine-coated coverslips. In analogy to PLL, Jurkat cells stop and scan the local environment after landing on glycine-coated coverslips, but do not migrate/crawl as on untreated surfaces (Supplementary Movies 3–5). Interestingly, a highly heterogeneous CD4-GFP signal was detected in most cells immobilized on glycine-coated coverslips for around 5 min since the first contact with the optical surface (Fig. 1c, d; Supplementary Fig. S1 and Supplementary Movie 2). Quantitative analysis of cell contacts demonstrated a significantly more heterogeneous distribution of CD4-GFP signal for Jurkat cells on glycine-coated coverslips compared to PLL (Fig. 1e, see "Methods"). Interference reflection microscopy images also indicated a more complex geometry of cells on glycine than on PLL, especially 5 min after landing (Supplementary Fig. S4). Intense spreading and partial surface flattening were observed for Raji (B) and RAW254.7 macrophage-like cells on PLL (Supplementary Fig. S5). On the contrary, a complex geometry of the contact was observed for Raji and RAW264.7 cells on glycine. Such difference was less pronounced for adherent cells (COS7 fibroblasts; Supplementary Fig. S5). We observed a narrower space between a cell body and optical surface for COS7 cells on PLL compared to glycine. This space is occupied by cell membrane protrusions and extensions. Altogether, our data demonstrate that glycine coating allows immobilization of cells close to the optical surface with improved and prolonged preservation of the cellular morphology, compared to the common PLL-coating. Such properties not only open access to improved live-cell imaging of cell surface morphology, but also to more convenient sample preparation, e.g., fixation, labeling,

etc. Of note, fixed and living Jurkat cells contacting glycine-coated coverslips show comparable morphologies (Supplementary Fig. S6).

**Improved viability and resting state of cells immobilized on glycine-coated coverslips.** We further examined the impact of our coating approach on the resting state and viability of immobilized T cells. PLL was reported to induce calcium response in T cells attached to the coated surface[68]. Our measurements confirmed that PLL induces calcium mobilization in Jurkat cells, albeit much less than antigenic stimulation mimicked by the coverslip coated with specific antibodies (anti-CD3ε) (Fig. 1f and Supplementary Fig. S7). In turn, the immobilization of Jurkat cells on glycine-coated coverslips stimulated delayed and reduced calcium response (Fig. 1f and Supplementary Fig. S7). Here, we employed the highly-sensitive, genetically encoded membrane-associated fluorescent calcium indicator GCaMP6f$_u$ and single-cell TIRF microscopy, which enables detection of weak and rapid signals in cells interacting with surfaces[69]. Similar results were observed in an experiment, in which a large number of cells (<1500) loaded with cytosolic calcium-sensitive fluorescent dye Fluo-4 was monitored over 15 min after their landing on coated coverslips (Supplementary Fig. S8a). A delayed and weaker calcium response was detected in Jurkat cells on glycine-coated coverslips compared to the cells on PLL or stimulating (anti-TCR) antibody. However, non-specific stimulation was observed in almost all Jurkat cells irrespective of the tested surface (Supplementary Fig. S8c). On the contrary, a half of primary CD4+ T cells isolated from human blood did not exhibit any calcium response on the glycine-coated surface during the whole period of the measurement (Supplementary Fig. S8c). No such protective effect was observed for primary CD4+ T cells on PLL- or antibody-coated coverslips. Nevertheless, stimulation of primary CD4+ T cells on PLL was much weaker compared to Jurkat cells on the same surface (Supplementary Fig. S8). Moreover, for longer cultivation times (>10 min) we found a significantly improved viability for cells incubated on glycine-coated coverslips compared to those on PLL-coated coverslips (Fig. 1g).

In summary, our findings indicate that coating of optical surfaces with glycine reduces the stress generated by the charged surface of, for example, PLL-coated coverslips. Importantly, the gel-like structure of the glycine coating better preserves the surface morphology of immobilized cells and enables quantitative analysis of the three-dimensional receptor distribution on the surface of resting T cells with SMLM.

**The three-dimensional SMLM method.** To capture the entire nanoscopic three-dimensional organization of receptors at the cell surface, we acquired image sequences with a highly inclined and laminated optical sheet (HILO) illumination, which enables single-molecule detection in cells much deeper than in TIRF microscopy (Fig. 2a). At the same time, it restricts the out-of-focus fluorescence from the remaining parts of a cell, thus improving the local signal-to-noise ratio[70].

*Intensity-based biplane SMLM imaging—dTRABI.* Previously, we reported the temporal, radial-aperture-based intensity estimation (TRABI) method in combination with a biplane detection as a powerful three-dimensional imaging tool with nanometer precision (<20 nm)[59,71]. In short, TRABI comprised fitting single-molecule spots by a Gaussian model with an invariant width, a subsequent independent photometric analysis of these spots and a final allocation of the axial coordinate based on a calibration curve. The photometric value was determined from the intensity

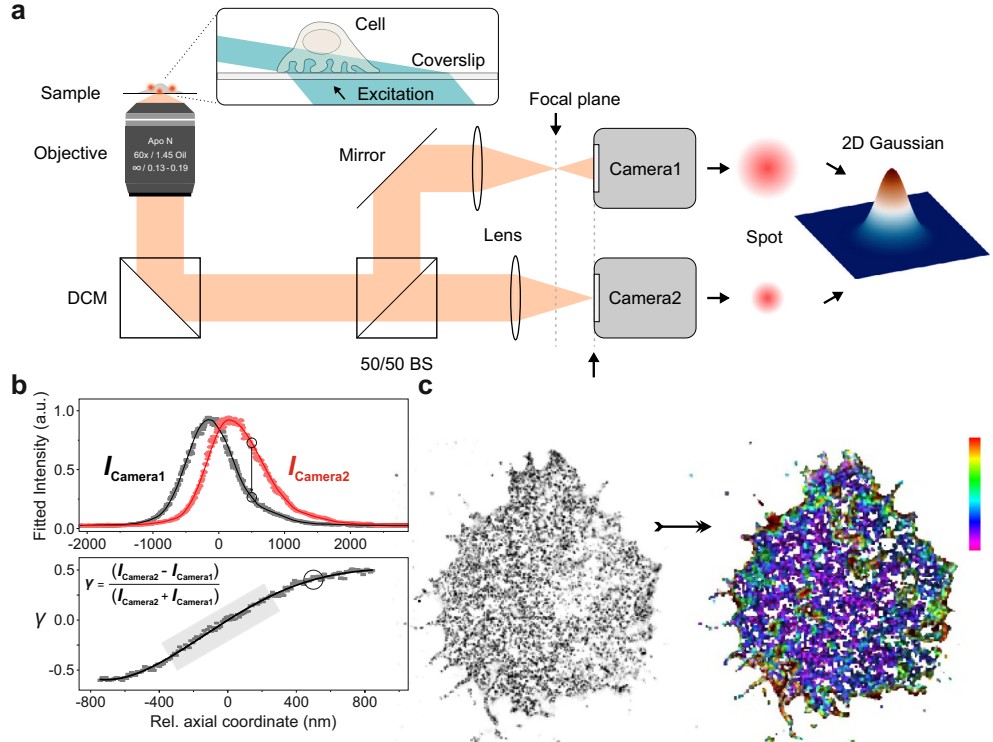

**Fig. 2 Principle of 3D super-resolution biplane imaging using dTRABI. a** Overview of the experimental setup applied to perform 3D dTRABI imaging of T cells. HILO illumination of the sample (blue beam; only shown in enlarged box) triggers fluorescence emission (orange), which is split by a 50/50 non-polarizing beamsplitter (50/50 BS) to acquire biplane images on two separate EM-CCD cameras. The respective imaging lenses are shifted along the optical axis to induce a relative defocus of the image detection on synchronized cameras. Spots, apparent in both detection planes, are fitted by a Gaussian with identically set FWHM. **b** Using a piezo stage, the focal plane was linearly moved through the sample plane while imaging a single-molecule surface under dSTORM conditions. Hereby, both cameras were synchronized. Fitting the raw PSFs by independent Gaussians with invariable FWHM yielded axially dependent single-molecule intensity curves (upper panel). The relative change of position of the imaging lens in the reflection path is mirrored by the relative shift of the respective intensity curve (indicated by circles). Data points were spline interpolated to guide the eye (solid lines). An axially precise calibration function $\gamma$ is derived directly from the raw intensities ($I_{Camera1}$, $I_{Camera2}$) of corresponding localizations from both cameras as $\gamma(z) = (I_{Camera2} - I_{Camera1})(I_{Camera2} + I_{Camera1})^{-1}$. The running median of the raw data (gray squares) is fitted with a high-order polynomial (black line) to generate the basis of the axial lookup table (lower panel). **c** A two-dimensional high-resolution data set is generated from both image stacks (transmission and reflection path) to create a three-dimensional dTRABI data set according to the calibration. Finally, the transmission localization set is used to render a high-resolution, axially color-coded image of the focused target structure. Scale bars, 5 μm.

obtained by the established Gaussian fitting procedure ($I_{Fit}$) and background-subtracted reference intensity of the spot ($I_R$), with $I_{Fit} < I_R$.

Here, we used a simplified but superior version of the TRABI method in which solely the intensity information obtained by the Gaussian fit ($I_{Fit}$) was used as metric for both channels in a biplane imaging scheme. Synchronized image stacks of the two channels (transmitted and reflected) generated by dividing the emitted fluorescence equally with a 50/50 beamsplitter, were analyzed by the same Gaussian function with an invariant width for every spot. Spots in the reflected channel were mapped to the corresponding spots in the transmitted channel by the nearest neighbor algorithm (Fig. 2b, see "Methods"). By omitting the standard photometric analysis[59], the computation time was reduced, and the allowed number of localizations per area and frame was significantly increased. Therefore, as demonstrated by Fourier Ring Correlation (FRC) of according data sets, we achieve an improved structural resolution while maintaining the same axial localization precision of better than 20 nm as with the original TRABI approach (Supplementary Figs. S9 and S10, see also "Methods")[72]. While this resolution-enhancement is more pronounced for some structures such as microtubules, and less distinct, but still apparent, for other structures such as CD4

clusters (Supplementary Fig. S10), there is an additional implication for molecular quantification approaches. We subsequently called the new method dTRABI (for *direct* TRABI; Fig. 2c).

*Fiducial-free drift and tilt correction.* To ensure optimal axial localization over an extended imaging time as well as accounting for subtle sample drift, we developed a fiducial-free and localization-based approach for its correction (see "Methods"). In short, the axial footprint of the entire structure is tracked over time, resulting in a spatio-temporal drift trace (Fig. 3a). This trace is fitted with a high-order polynomial, which serves as the correction term for the raw localizations by linearization. We assumed that in a thin sample layer like the plasma membrane, the spatio-temporal distribution of active photo-switches that reside in their on-state is constant over time. The selection of appropriate regions in axially more extended samples can include structures, where the local z-dimension is restricted. Most data sets that were used to conclude the results in the following paragraphs showed axial drift with different orders of magnitude and non-linearity, which could be accounted for by the described correction approach. Based on the calculated axial localization precision, this approach yields at least a similar accuracy,

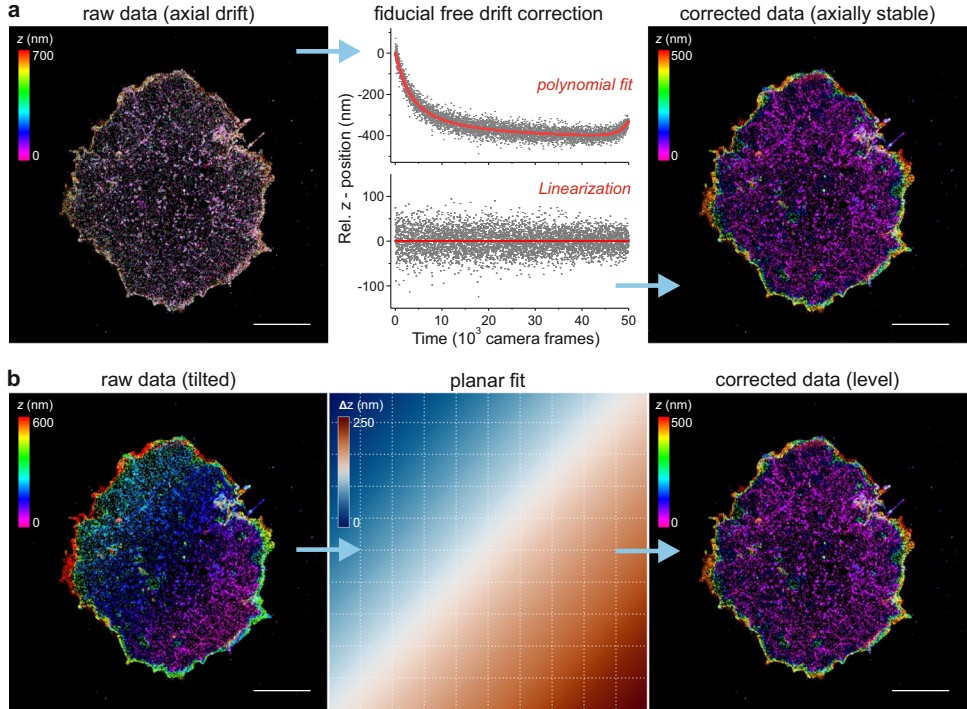

**Fig. 3 Fiducial free drift and tilt correction of dTRABI data.** CD4 WT on Jurkat cells is shown, where the membrane was labeled using Alexa Fluor 647-conjugated primary antibody (OKT4). **a** Principle of the fiducial-free correction of axial drift. Left: representative, color-coded high-resolution dTRABI image of an axially unstable sample. Due to the temporal change of the axial coordinate, the resulting image does not exhibit axially distinct features. Middle: by extracting the axial localization distribution of the thin membrane layer per frame and tracking it over the entire stack, a spatio-temporal drift trace can be plotted (top). Fitting of the raw data (gray squares) by a high-order polynomial (red line) allows the temporal linearization of the localization data, leading to a stable axial mean value over time (bottom). Right: re-rendering of the drift-corrected localization data reveals a color-coded dTRABI image exhibiting distinct, high-density clusters of CD4 below another, disperse, layer. **b** Principle of the fiducial-free correction of axial tilt. Left: representative, color-coded high-resolution dTRABI image of an axially tilted sample. Middle: the axial tilt of the sample is extrapolated by fitting a plane to the raw image. Afterwards, the data are linearized by subtracting the local plane-value from the raw localization. Right: the tilt-corrected image exhibits a homogeneous color-code in the lowest data layer, indicating no residual axial tilt. Exemplary drift and tilt were simulated for illustration but based on real data (the displayed cell displayed both significant drift and tilt). Right columns represent experimentally corrected real data on which the drift and tilt were projected. For all experimental data, axial drift correction was performed prior to a tilt test and correction. Scale bars, 5 μm.

compared to fiducial-based methods, but without any additional interference with the sample. However, an additional linear axial tilt was observable in some data sets, most likely due to minor imperfections of the sample holder. Since we assumed the layer closest to the coverslip to be axially flat on the whole-cell scale, we fitted an inclined plane to the raw image, thereby determining its gradient and thus enabling the linearization of the raw localization data (see "Methods" and Fig. 3b).

*Quantitative axial localization analysis.* For the analysis of the axial receptor distribution, localization files were loaded and processed in ImageJ/Fiji with custom written scripts (see "Methods")[73,74]. The analysis comprised three steps: generation of quantitative image stacks, segmentation in regions of interest (ROIs) and axial quantification of individual ROIs. (1) A quantitative 3D image stack ($z$-stack) was generated with 20 nm pixel size in $x$, $y$ and $z$. (2) These stacks were then segmented by automatically generating multiple ROIs with an area of 3–4 μm² within the interior of the cell (Supplementary Fig. S11). (3) Then, the axial distribution of localizations was analyzed by counting all localizations per ROI along the $z$-stack. For each ROI, the resulting distribution was fitted to a superposition of two Gaussians (Supplementary Fig. S11). By analyzing the properties of these fits, we acquired parameters that provide quantitative information about the axial distribution of surface receptors (see "Methods").

**Imaging CD45 nanotopography using the dTRABI approach.** To demonstrate the applicability of dTRABI, we labeled CD45 receptors, which are highly expressed on the surface of T cells (>100.000 copies/cell; ref. [75]) with Alexa Fluor 647-conjugated antibody (MEM-28) and imaged as described above. Cells were immobilized on glycine-coated coverslips to preserve cell surface morphology. Resulting three-dimensional images with color-coded $z$-axis represent a footprint of a cell on a coverslip (Fig. 4a). The apparent optical depth, which varies between cells (300–650 nm), depends on the available structure in the individual field of view. CD45-labeled cells exhibited a variety of features that can extend far from the coverslip. However, the entire extent of the basal membrane resting on the surface was captured in all displayed and analyzed cells. Figure 4b shows the three-dimensional image of CD45 which indicates a complex morphology of the cell surface. Magnified images and the corresponding $xz$-projections (Fig. 4c) demonstrate the ability of dTRABI to visualize large membrane protrusions at the edge of the cell (ROIs 1 and 3; blue arrowheads), as well as more subtle membrane extensions toward the exterior (ROIs 1–4; red arrowheads) and interior (ROIs 3 and 5; green arrowheads) of the cell.

**The origin of CD4 microclusters on the surface of unstimulated T cells.** Next, we aimed at examining the origin of previously reported receptor (signaling) microclusters[1,4,5]. We previously

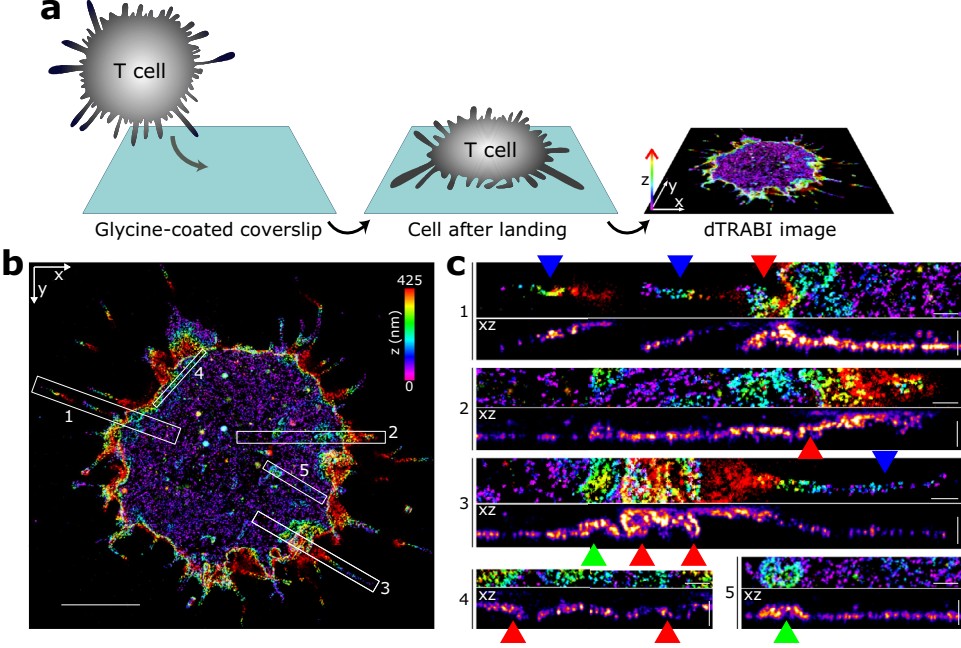

**Fig. 4 Cell surface receptor nanotopography visualized by 3D TRABI imaging. a** Schematic illustration of sample preparation for a receptor nanotopography imaging using dTRABI. A cell in suspension is first immobilized on a glycine-coated coverslip, fixed and imaged using dTRABI approach. The resulting three-dimensional dTRABI image represents a footprint of a cell on the optical surface with the color-coded axial position of localizations (right panel). **b** Three-dimensional dTRABI image of T-cell surface receptor CD45 with selected ROIs exhibiting a broad z-distribution of receptor localizations. CD45 was labeled with Alexa Fluor 647-conjugated primary antibody (MEM-28). **c** Magnified x–y and x–z projections of ROIs 1–5 as in **b**. Blue arrowheads point to microscopic membrane protrusions at the cell edges, red arrowheads to folded nanoscopic protrusions under the cell body and green arrowhead to membrane invagination. Scale bars, 5 μm in **b** and 500 nm in **c**.

showed that CD4 receptors accumulate in high-density regions on unstimulated T cells using two-dimensional SR imaging[28], while the three-dimensional organization of these clusters remained unclear. First, we confirmed that CD4 accumulates in high-density regions on T cells immobilized on glycine-coated coverslips using 2D dSTORM and Alexa Fluor 647-conjugated antibody (OKT-4) (Supplementary Fig. S12). Next, high-resolution three-dimensional maps of the CD4 surface distribution were acquired by analyzing the same data sets with the dTRABI approach (Fig. 5a). The high-precision, nanoscopic three-dimensional view of wild-type CD4 indicates that the receptor is clustered and preferentially localized to one topographical level of the membrane (Fig. 5a). This is also demonstrated by plotting the relative z-position of CD4 molecules in the central area, which avoids cell edges (Fig. 5b). On the contrary, a broad surface distribution is evident from the three-dimensional view of a CD4 variant, which cannot be palmitoylated and was previously shown to exhibit random distribution in such unstimulated T cells (CD4 CS1; Fig. 5c, d; ref. [28]). Though these plots provide useful information about axial distribution of receptors in individual cells, their utility for the global distribution analysis is limited by a variation in surface complexity of individual cells. Therefore, the propensity of CD4 molecules to accumulate at specific topographical levels was analyzed in $2 \times 2\,\mu m^2$ square ROIs selected to cover the cell-coverslip contact area over all tested cells (in total 846 CD4 WT ROIs in 21 cells and 1044 CD4 CS1 ROIs in 18 cells; Supplementary Fig. S11 and Table 1). As mentioned above, for quantitative analyses, the distribution of relative z-positions for localizations in ROIs were fitted with a sum of two Gaussians (bi-Gaussian, Fig. 5e). From these fits, three parameters were derived, i.e., the spread of the bi-Gaussian termed z-distribution width ($z_w$) (Fig. 5f, g), distance of its mean values or peak-to-peak distance (p-p) (Supplementary Fig. S13) and the difference of the widths of

**Table 1 Overview of distribution parameters derived from axial receptor analysis.**

|  |  | WT | CS1 | CD45 |
|---|---|---|---|---|
| N cells |  | 24 | 18 | 13 |
| N ROIs |  | 846 | 1044 | 305 |
| $z_w$ | $\mu_1$ (nm) | 120 | 141 | 118 |
|  | $\sigma_1$ (nm) | 25 | 27 | 20 |
|  | range$_1$ (nm) | 90-150 | 110-173 | 94-145 |
|  | $\mu_2$ (nm) | - | 247 | 263 |
|  | $\sigma_2$ (nm) | - | 83 | 117 |
|  | range$_2$ (nm) | - | 149-345 | 126-400 |
| p-p | $\tau$ (nm) | 23 | 36 | 121 |
| $\Delta_{FWHM}$ | $\mu$ (nm) | 57 | 88 | 63 |
|  | $\sigma$ (nm) | 15 | 28 | 22 |
|  | range (nm) | 40-75 | 55-120 | 37-89 |

N is the number of cells or ROIs. $z_w$ = z-distribution width, p-p = peak-to-peak distance, $\Delta_{FWHM}$ = width difference; μ is the mean and σ is the standard deviation of the histogram, range = μ ± FWHM/2 with FWHM = 2.355σ; τ is the mean of the exponential distribution. The raw data are shown in Fig. 5 and Supplementary Fig. S13.

the two Gaussians ($\Delta_{FWHM}$) (Supplementary Fig. S13). If $z_w$ describes the overall broadness of the axial receptor distribution, p-p indicates how well separated the individual populations are, described by the two Gaussians. $\Delta_{FWHM}$ informs how much broader the axial distribution of the second population is, which potentially covers receptors at the plasma membrane base. Together, these parameters enabled a direct comparison of receptors (CD4 vs. CD45) and their variants (CD4 WT vs. CD4 CS).

By compiling the $z_w$ values of all ROIs, a histogram was obtained (Fig. 5f) and further analyzed ("Methods"). For wild-type CD4, we calculated an axial mean value of 120 nm and

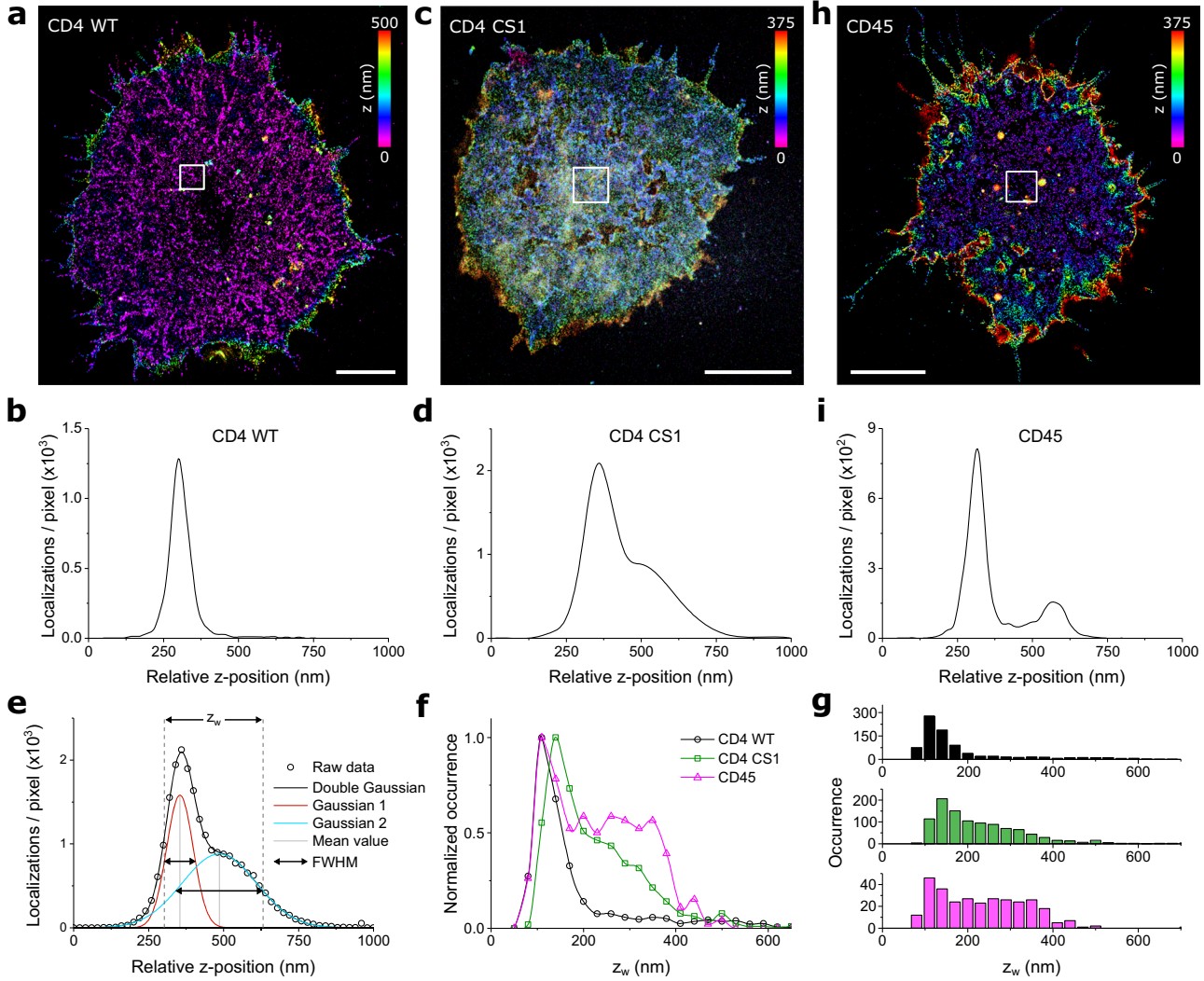

**Fig. 5 CD4 clusters represent the receptor accumulation at the tips of membrane protrusions. a** Representative three-dimensional dTRABI image of CD4 WT at the surface of Jurkat cell immobilized on a glycine-coated coverslip. **b** The axial distribution of CD4 WT localizations of the 4 µm² region of interest (ROI) of the cell in **a**. **c** Representative 3D dTRABI image of non-palmitoylatable CD4 CS1 mutant at the surface of Jurkat cell immobilized on a glycine-coated coverslip. **d** The axial distribution of CD4 CS1 localizations of the 4 µm² ROI of the cell in **c**. **e-g** Quantitative analysis of the receptor axial (z-axis) distribution on the surface of T cells. Receptors were analyzed by using a bi-Gaussian fit to the axial distribution of localizations for each ROI as in **b**, **d** and **i** (see Supplementary Fig. S11 for more examples and Supplementary Data for the details) and the FWHM range of the two Gaussian functions represent the z-distribution width ($z_w$) as depicted in **e**. Black circles in **e** represent the axial distribution of receptor localizations for a selected ROI, black line the bi-Gaussian fit, which is the sum of two Gaussians as depicted in red and blue, dashed lines in gray depict $z_w$ and lines in light gray depict mean values of the Gaussian distributions. The graphs in **f** and **g** represent histograms of $z_w$ obtained from 21 CD4 WT cells with 846 ROIs (black), 18 CD4 CS1 cells with 1044 ROIs (green) and 13 CD45 WT cells with 305 ROIs (magenta). The histograms in **f** and **g** show relative and absolute occurrence, respectively. Data points in **b**, **d**, **f** and **i** were spline interpolated to guide the eye. **h** Representative 3D dTRABI image of CD45 at the surface of Jurkat cell immobilized on a glycine-coated coverslip. Color-bars in the upper right corner of **a**, **c** and **h** indicate the axial position of the localizations in the image. **i** The axial distribution of CD45 localizations of the 4 µm² ROI of the cell in **h**. CD4 variants and CD45 were stained with directly labeled primary antibody as in Figs. 3 and 4, respectively. Scale bars in **a** and **c** and **h**, 5 µm.

a standard deviation (s.d.) of 25 nm (cf. Table 1), signifying a strong tendency of wild-type CD4 to accumulate at one specific topographical level in resting T cells (Fig. 5f, g). On the contrary, a lack of palmitoylation in CD4 CS1 variant caused much broader distribution of this receptor on the T-cell surface, with $z_w$ predominantly at 141 nm ± 27 nm (mean ± s.d.) and a smaller, yet broader fraction at 247 ± 83 nm (mean ± s.d.). Furthermore, the axial analysis of localizations revealed a more pronounced mean value of $\Delta_{FWHM}$ for CD4 CS1 (88 nm) than for CD4 WT (57 nm) (Supplementary Fig. S13c and Table 1). The different axial distribution of the two CD4 variants on Jurkat cells was further

confirmed by averaging the localizations for all cells tested (21 CD4 WT and 18 CD4 CS1 cells; Supplementary Fig. S14).

The specificity of CD4 receptor clusters was further emphasized by a less constrained lateral and axial distribution of CD45 in T cells imaged using dTRABI (Fig. 5h, i; 13 cells and 305 ROIs were analyzed). Quantitative analyses were performed consistent with the CD4 variants and demonstrated a broader axial distribution of CD45 receptors than CD4 WT (Fig. 5f, g). We found the $z_w$ values to be similarly distributed at 118 ± 20 nm (mean ± s.d.), but in contrast to CD4 WT, CD45 had an additional large fraction of $z_w$ at 263 ± 117 nm (mean ± s.d.).

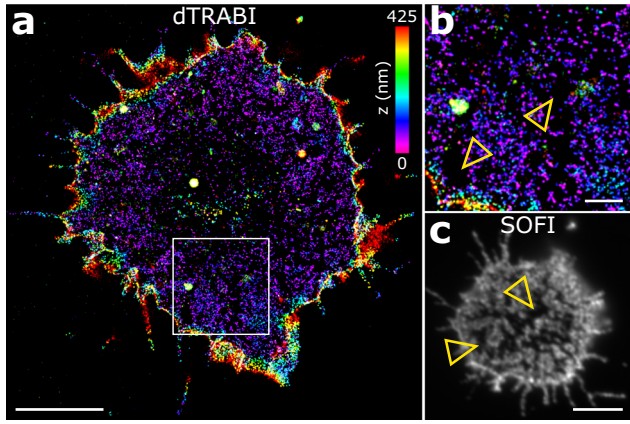

**Fig. 6 CD45 exclusion zones on the surface of resting Jurkat cells. a** Three-dimensional dTRABI image of CD45 (labeled with antibody as in Fig. 4) on Jurkat cell immobilized on a glycine-coated coverslip. **b** Magnified ROI as in **a** with indicated areas lacking CD45 localizations (yellow arrowheads). **c** 2D SOFI image of CD45 on Jurkat cell immobilized on a glycine-coated coverslip. CD45 was labeled using Alexa Fluor 647-conjugated MEM-28 antibody. Yellow arrowheads indicate cell surface areas lacking CD45 signal. Scale bars, **a** 5 μm, **b** 1 μm and **c** 5 μm.

This broad CD45 distribution was further noticeable when analyzing the peak-to-peak distance (p-p) of the fitted bimodal distribution to the relative $z$-position of the receptor (Supplementary Fig. S13b, see "Methods"). The obtained p-p values were exponentially distributed and the mean values for CD4 WT and CS1 were 23 and 36 nm, respectively, whereas the mean value for CD45 was 121 nm. More examples of three-dimensional dTRABI images and their quantitative analysis for all three receptors are shown in Supplementary Figs. S15 and S11, respectively.

**Segregation of CD45 from CD4 clusters in non-stimulated T cells.** When analyzing dTRABI images, we noted areas with very few CD45 localizations (Fig. 6a, b; yellow arrowheads). Similarly, CD45 exclusion zones were detectable in two-dimensional SOFI images of antibody-labeled T cells (Fig. 6c). The narrow shape of these exclusion zones was reminiscent of the CD4 distribution observed on T cells imaged by the dTRABI approach (Fig. 5a and Supplementary Fig. S15). We thus wondered whether the CD45 exclusion zones represent areas with CD4 accumulation. T cells expressing CD4 WT fused to photo-switchable protein mEos2 (ref. [28]) were labeled with specific anti-CD45 antibodies and analyzed using two-color SMLM. Our data indicate that CD4 and CD45 are essentially segregated into two separate zones on the surface of non-stimulated Jurkat cells immobilized on glycine-coated coverslips (Fig. 7a, b). Since CD4 clusters to one topographical level near the cell-coverslip contact sites (Fig. 5a, b, f), we suspect that CD4 preferentially accumulates at the tips of membrane protrusions (e.g., microvilli) and CD45 to the shaft of these structures and their base at the plasma membrane (Fig. 7c, f). Indeed, we observed the accumulation of CD4 at the tips of large T-cell membrane protrusions in several cells (Fig. 7d, e). In turn, the shaft of protrusions was extensively covered with CD45 signal. Often, CD45 was essentially segregated from CD4 in these membrane structures (Fig. 7e). These data are in agreement with recently reported observations that CD45 segregates from microvilli tips in resting and activated T cells[76,77].

## Discussion

In this work, we have introduced a straightforward approach for molecular mapping of the membrane receptor topography with nanometer precision. The method combines the use of glycine-coated optical surfaces, which helps to preserve the membrane morphology of immobilized cells with an advanced and simplified dTRABI algorithm for quantitative, near isotropic three-dimensional SMLM imaging. Using directly labeled antibodies and dTRABI processing of biplane SMLM data, we achieved 10–20 nm localization precision in all spatial directions.

Our principal task was to determine the distribution of signaling receptors on the complex T-cell surface. Standard methods of sample preparation (e.g., on PLL-coated coverslips) can rapidly damage the fragile surface morphology of studied cells, as indicated by our data showing that the cell surface complexity is diminished within a couple of minutes of the first contact with a PLL-coated coverslip. Several materials were previously developed to prevent cell deformation during their immobilization on coverslips, for example, Matrigel and synthetic hydrogels. Matrigel enables three-dimensional cell cultures, but it is derived from the extracellular matrix of mouse tumors and exhibits auto-fluorescence. On the contrary, fluorescence properties of synthetic hydrogels (e.g., CyGEL Sustain) can be controlled and the cells prepared in hydrogels display numerous protrusions on their surface, including nanoscopic structures that require SR microscopy for their detection[78]. However, TIRF and HILO sample illumination techniques which can reduce the out-of-focus background signal and, thus, improve the quality of SMLM images, enable fluorescence signal detection solely in the vicinity of the optical surface[70,79]. Hydrogels do not allow efficient positioning of immobilized cells at the optical glass.

Here, we developed and analyzed glycine coating of coverslips, which better preserves cell membrane morphology compared to PLL. The coating of coverslips with a narrow layer of gel-like glycine improves the stability of membrane protrusions and facilitates the immobilization and positioning of cells close to the optical surface. Importantly, cells immobilized on glycine exhibit improved viability and reduced non-specific stimulation compared to PLL-immobilized cells. Of note, the preparation of glycine-coated coverslips is straightforward, fast (<30 min) and inexpensive. These properties qualify glycine coating of coverslips for a range of advanced imaging approaches of cells, including high-resolution three-dimensional dTRABI mapping of cell surface molecules.

Membrane imaging with two-dimensional methods leads to simplifications and, potentially, misinterpretations. The availability of SR microscopy capable of resolving nanoscopic three-dimensional structures is currently limited to specialized nanophotonics laboratories. To enable such imaging to a broader spectrum of scientists, we have developed the dTRABI algorithm for a quantitative analysis of the axial distribution of fluorescent molecules. dTRABI allows for faster and computationally more efficient processing of SMLM data, while achieving significantly higher structural resolution, than that found with our original TRABI approach. Importantly, dTRABI can be used with any localization software that supports fixed Gaussian fitting in combination with a biplane detection scheme. In contrast to classical biplane imaging that compares the width of spots evaluated by free Gaussian PSF fitting, dTRABI employs fixed-width Gaussian fit which facilitates an improved localization precision. The ability of dTRABI to localize fluorescent molecules on complex, irregular, three-dimensional cellular structures was highlighted on a Jurkat cell that was not included in the quantitative analysis of an axial receptor distribution (Supplementary Fig. S16). Complex membrane structures formed by this dying cell were effectively labeled and visualized using the dTRABI approach.

dTRABI offers a tool for mapping localization of molecules within ~1 μm of the optical surface. Such working space provides

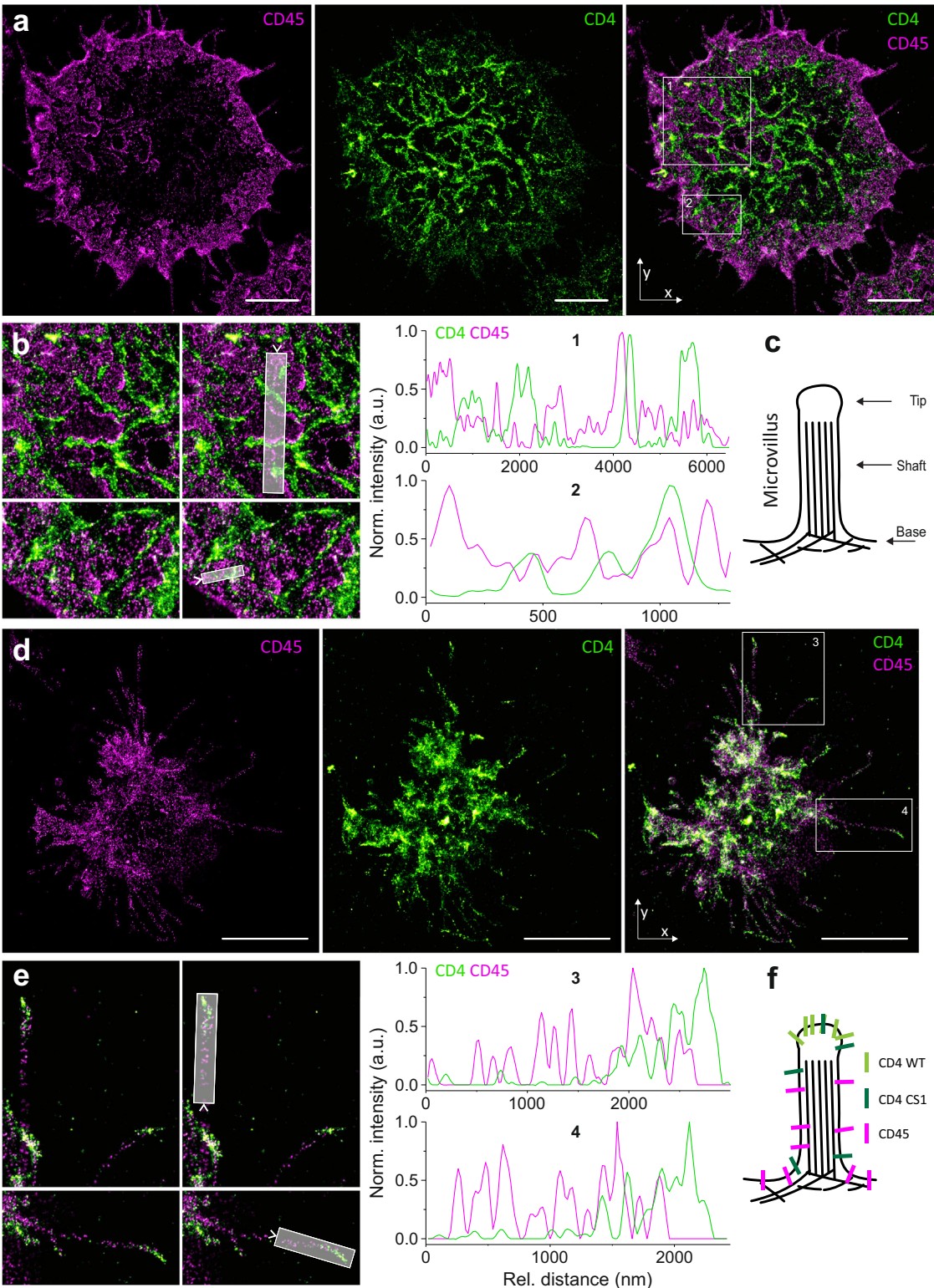

a sufficient axial signal penetration to visualize the eukaryotic plasma membrane with its nanotopography. Under the conditions used in our experiments, the plasma membrane protrusions were positioned 100–400 nm from the optical surface (Fig. 5). Thus, molecules that do not localize to the protrusions can be monitored as well. This was facilitated by the folding of membrane protrusions, the length of which often exceeded the axial depth of our method, under the cell body (Fig. 4 and

Supplementary Fig. S17a, b). Evidently, a partial loss of molecular localizations in the areas where the base of the plasma membrane is positioned outside of the optical limit of this method is caused by a diverse positioning of cells and intensive bending of cell-glass contacts (Supplementary Figs. S15 and S17c, d).

The quantitative character of dTRABI data also enabled a detailed analysis of axial receptor distribution, which we modeled by fitting a sum of two Gaussian functions (bi-Gaussian) to the

**Fig. 7 Nanoscopic segregation of CD4 WT and CD45 on the surface of resting Jurkat cells. a** Representative two-dimensional images of Jurkat cell surface sequentially analyzed for CD45 (magenta; left panel) and CD4 WT (green; middle panel) by SMLM. The right panel represents an overlaid image. CD4 was visualized as mEos2 fusion protein (PALM) after transient transfection of cells, and the surface CD45 was labeled using Alexa Fluor 647-conjugated MEM-28 antibody (dSTORM). ROIs 1-2 were zoomed to indicate details of proteins' distribution (right side). **b** Intensity line-profiles were measured along the transparent gray regions indicated in ROIs 1–2 (as in **a**). Green line represents CD4 WT and magenta CD45 signals. **c** Schematic illustration of microvillus with indicated structural segments: tip, shaft and the basis. **d** Two-dimensional SMLM images of a selected Jurkat cell captured during its association with the optical surface which was sequentially analyzed for CD45 and CD4 WT (as in **a**). The accumulation of CD4 WT on the tips of large membrane protrusions was observed in 10 out of 32 imaged Jurkat cells. ROIs 3–4 were zoomed to show the details of membrane protrusions with accumulated CD4 WT on their tips (right side). **e** Intensity line-profiles were measured along the transparent gray regions indicated in ROIs 3–4. The arrows indicate the onset of line-profiles. Scale bars, 5 μm. Images from three independent experiments are shown ($n = 32$). **f** Schematic illustration of the organization of CD4 WT, CD4 CS1 and CD45 on the protrusions as indicated by the nanoscopy.

axial localization data. Although CD4 distribution can be modeled by a single Gaussian to a high degree of confidence, a bi-Gaussian is even more accurate and accounts for a slight asymmetric organization of receptors. In contrast, non-palmitoylatable CD4 CS1 mutant does not follow a mono-Gaussian distribution and the bi-Gaussian is essential. The analysis allowed us to derive quantitative parameters describing axial receptor distribution in detail, i.e., $z_w$, p-p and $\Delta_{FWHM}$. All these analytical approaches emphasized the narrow and symmetrical distribution of CD4 WT, while CD4 CS1 and CD45 show significantly broader distributions.

To highlight the effectiveness of our new imaging approach, we explored two important biological questions: (1) what is the origin of receptor microclusters in T cells, and (2) how does membrane morphology affect segregation of signaling receptors into different areas of the plasma membrane? Both questions cannot be studied by previously reported protocols. Using quantitative SOFI analysis, we have previously shown that CD4 accumulates in high-density regions (clusters) in unstimulated T cells by a process that depends on its post-translational lipid modification, palmitoylation[28]. However, as in many other cases, these data represented two-dimensional projections of receptor localizations on the complex T-cell surface. Using dTRABI imaging of unstimulated Jurkat cells immobilized on glycine-coated coverslips, we demonstrate that CD4 accumulates on the tips of microvilli (or membrane ruffles). Thus, CD4 clusters, and potentially other clusters, represent receptors (or other molecules) trapped at the tips of membrane protrusions, stressing that intact cell surface morphology is essential for the proper spatial representation of signaling molecules. Our data agree with recently reported clustering of CD4 on the tips of T-cell microvilli[80] but provide more detailed and quantitative insight into the CD4 distribution to these structures. Importantly, the mutant CD4 CS1 variant which cannot be palmitoylated distributed more randomly over the complex surface of T cells. Targeting to or stabilizing proteins on membrane protrusions thus may be another, previously unreported role of protein palmitoylation[81].

Several molecules, including CD45, were shown to segregate from signaling microclusters upon T-cell activation (e.g., refs. [82,83]). Using dTRABI and glycine-coated coverslips, we demonstrated that CD45 distributes over a broader axial spectrum than CD4. Our data thus indicate that CD45 segregates from signaling molecules by localizing to the base of the plasma membrane. A similar observation was recently reported by the group of Klaus Ley[76]. They further demonstrate that such localization is determined by the transmembrane domain of CD45.

In summary, we provide a new and undemanding workflow to study nanotopography of receptors at the cell surface. We highlight the importance of appropriate sample preparation for the imaging of three-dimensional structures. The improved algorithm of dTRABI exemplifies an easily implemented and high-quality method to generate three-dimensional SMLM data. Finally, we demonstrate the applicability of the workflow by answering two critical questions related to the localization of surface receptors on lymphocytes. Even though the method was used here for human receptors, it can be implemented for the molecular characterization of the surface of other organisms (e.g., yeast, plants) but also for complex nanomaterials.

## Methods

**Cell culture and transfection.** Jurkat cells (clone E6-1; ATCC), their CD4-knock-out variants (Jurkat CD4-KO cells) and Raji B cells (CCl-86, ATCC) were grown in RPMI-1640 medium (Fischer Scientific) supplemented with 10% fetal bovine serum (Fischer Scientific), Non-essential amino-acids (Fischer Scientific) and 10 mM HEPES under controlled conditions in a humidified incubator at 37 °C, and 5% $CO_2$ (Eppendorf). Rhesus monkey COS-7 fibroblastic cells (CRL-1651, ATCC) and mouse RAW264.7 macrophage cell line (TIB-71, ATCC) were grown in DMEM (High-Glucose, Sigma-Aldrich) supplemented with 10% FBS under controlled conditions in a humidified incubator at 37 °C, and 5% $CO_2$. Jurkat CD4-KO cells were derived from wild-type Jurkat cells using CRISPR/Cas9 technology as described[84]. All cell lines were regularly tested for morphology and mycoplasma infection.

For the expression of exogenous proteins, Jurkat cells, their CD4 KO variant, Raji and RAW264.7 cells were transiently transfected with plasmid DNA using Neon® transfection system (Thermo Fisher Scientific) according to the manufacturer's instructions. Briefly, 1 μg of vector DNA was used per 200,000 cells in 0.5 ml culture medium. The instrument settings were: (1) for Jurkat cells and their CD4 KO variant 3 pulses, each 1350 V for 10 ms, (2) for RAW264.7 cells 1 pulse 1680 V for 20 ms, and (3) for Raji cells 1 pulse 1350 V for 30 ms. For transfection of COS-7, cells were seeded at $2.5 \times 10^5$ per 25 mm coverslips coated with glycine or PLL a day before transfection. Cells were transfected using TransIT-X2® Dynamic Delivery System (Mirus) using 0.5 μg of plasmid according to the manufacturer's recommendations. Transfected cells were used 12–36 h since the transfection.

**Isolation of primary CD4+ T cells.** Lymphocytes were pre-purified from buffy coats using Ficoll-Paque separation protocol[85]. Afterwards, primary CD4+ T cells were isolated using magnetic beads in Human CD4+ T-Cell Isolation Kit (Miltenyi Biotech, 130-096-533) and AutoMACS system (Miltenyi Biotech) according the manufacturer instructions. Cell purity (84–86%) was verified using immuno-fluorescence and flow cytometry. Isolated cells were grown in RPMI-1640 medium (Fischer Scientific) supplemented with 10% fetal bovine serum (Fischer Scientific), non-essential amino-acids (Fischer Scientific) and 10 mM HEPES under controlled conditions in a humidified incubator at 37 °C, and 5% $CO_2$ (Eppendorf).

**Sample preparation for SR microscopy**

*Cleaning and handling of coverslips and solutions for microscopy.* High-precision microscopy coverslips (round, 25 mm in diameter; Marienfeld) in Teflon holders (Wash-N-Dry Coverslip Rack; Diversified Biotech Inc.) were cleaned by incubation at 56 °C overnight in 2% Hellmanex III (Hellma Analytics) dissolved in ultrapure Milli-Q® water (Millipore) followed by 30 min sonication in a heated sonication bath. Several washes in ultrapure water and one additional sonication step were used to remove all traces of Hellmanex III components. Cleaned coverslips were stored in ultrapure water to avoid drying and contamination with particles of a dust from air.

All other glassware was regularly treated with piranha solution (3 parts of 30% hydrogen peroxide and 7 parts of concentrated sulfuric acid) for 45 min followed by several washes with ultrapure water. All solutions were made from concentrated stocks stored in glass containers, diluted with ultrapure water and filtered using syringe filters with 0.22 μm pores (TPP) into piranha-treated glassware.

*Glycine coating of coverslips.* Ultraclean coverslips were coated by applying 0.5 ml of 2 M glycine solution and incubation for 20 min at room temperature in the laminar flow box. Afterwards, the liquid phase was aspirated, and the hydrogel was washed with ultrapure water. Coverslips coated with glycine hydrogel were used immediately for immobilization of cells. The hydrogel-coated coverslips can be stored at 4 °C covered with a layer of ultrapure water for several days.

*PLL coating of coverslips.* Ultraclean coverslips were coated by applying 0.5 ml of 0.01% (w/v) PLL solution in ultrapure water and incubation for 20 min at room temperature in the laminar flow box. Unbound PLL was removed by aspiration of the liquid and a single wash with ultrapure water. Dried, PLL-coated coverslips can be stored for several weeks. For SR imaging, freshly prepared PLL-coated coverslips were used exclusively.

*Immobilization and fixation of cells for SR imaging.* Jurkat cells express varying levels of CD4 which are changing when cells are grown in culture. Therefore, we used transiently transfected Jurkat-CD4KO cells with reintroduced CD4-GFP to study CD4 surface distribution with dTRABI. Cells expressing similar levels of CD4-GFP were selected for imaging and data processing.

For SR imaging, after centrifugation for 3 min at $500 \times g$ (room temperature), cells were resuspended in pre-warmed PBS (made from 10x stock, Fischer Scientific) and seeded on glycine-coated coverslips immediately after the aspiration of a coating liquid. Cells were enabled to land on the coated optical surface for 10 min at 37 °C in the CO₂ incubator. Afterwards, cells were fixed with pre-warmed 4% paraformaldehyde (Electron Microscopy Sciences) containing 2% saccharose in PBS for 45 min at room temperature and the process was stopped with 50 mM NH₄Cl (Sigma-Aldrich) in PBS and three rounds of washing with PBS.

*Immunofluorescence.* For labeling, cells were first incubated with 5% BSA in PBS (Blocking solution) for 1 h at the ambient temperature to prevent non-specific binding of antibodies. Immunostaining of specific receptors was performed by incubating cells overnight with Alexa Fluor 647-conjugated primary antibodies (human CD4: OKT4, dilution: 1:100, source: Biolegend; human CD45: MEM-28, dilution: 1:2000, source: ExBio) diluted in Blocking solution. The process was performed at the ambient temperature in a dark humid chamber to avoid drying of the solutions and exposure to light. After removing of redundant antibodies and 3 times washing with PBS for 5 min, the cells were post-fixed with 4% PFA and 0.1% glutaraldehyde in PBS for 5 min and washed 5 times with PBS.

The identical protocol was applied for the staining of CD45 on Jurkat CD4 KO cells transfected with pXJ41-CD4-mEos2 plasmid for 2-color 2D SMLM analysis.

**Single-molecule localization microscopy.** Three-dimensional SMLM experiments for dTRABI analysis were performed on a home-built wide-field setup, which is described elsewhere in detail[59]. Raw image stacks were analyzed with rapidSTORM 3.2 (ref. [86]). Herein, the FWHM was set to 300 nm as an invariant parameter. Furthermore, the lower intensity threshold was set to 500 photons and the fit window radius to 1200 nm. All other fit parameters were kept from the default settings in rapidSTORM 3.2. Linear lateral drift correction was applied manually by spatio-temporally aligning distinct structures to themselves. This was facilitated by color-coding of the temporal coordinate with the built-in tool.

Two-color SMLM experiments were performed on a Nikon Eclipse Ti microscope, which is specified elsewhere in detail[45]. Contrary to the case of dTRABI analysis, the FWHM was set as a free fit parameter, but in the limits of 275–750 nm, both for Alexa Fluor 647 antibody and mEos2 fusion protein localizations. All other parameters were kept consistent with the previous experiments. Prior to imaging, a glass surface with Tetraspeck beads (Thermo Scientific) was imaged with alternating 561 nm and 647 nm excitation to create a nanometer precise map to allow the correction of chromatic shift. The 561 nm excitation channel was then mapped onto the 647 nm after the localization step.

Prior to the acquisition, we briefly irradiated the whole-cell in epifluorescence to further minimize out-of-focus contributions by transferring the majority of dyes into the non-fluorescent dark state. In all experiments, the length of the acquisition was set to capture the majority of emitters, i.e., imaging was concluded when only a very minor number of active emitters were detectable. A comparison of dSTORM image, Gauss-convolved rendering of dSTORM image and diffraction-limited TIRF image acquired prior to dSTORM imaging demonstrates that the employed SR method sufficiently reproduces native receptor distribution on T cells (Supplementary Fig. S18). Typical acquisition lengths were 60,000–120,000 frames for Alexa Fluor 647 channel and 30,000–60,000 frames for mEos2 channel, where integration times were set to 20 and 16 ms in the single- and dual-color cases, respectively. Hereby, mEos2 was excited at 561 nm and activated with 405 nm. The activation power density was increased over the time to create an almost constant signal density.

**Intensity-based biplane imaging—dTRABI**

*The principle of dTRABI.* We previously reported TRABI-based biplane (BP-TRABI; ref. [59]), which utilizes the photometric ratios $P$ of the molecules for the image reconstruction. $P$ was calculated as the quotient of the fit intensity ($I_{Fit}$) and reference intensity with the vast majority of photons of the spot (referred to as $I_R$ in

the following), where $I_{Fit}$ was derived from a PSF fit with fixed width, which was set in rapidSTORM software. For BP-TRABI, the photometric ratio of both planes ($P_{1,2}$) was calculated according to:

$$P_{1,2}(z) = \frac{P_1 - P_2}{P_1 + P_2} \tag{1}$$

with $P_1$ and $P_2$ being the photometric ratio of the individual planes.

The quality of the resulting axial coordinate depended on the precision of the TRABI intensity measurement. In order to exclude any interference from neighboring emitters as well as temporal overlap with other fluorophores, a set of rigorous exclusion criteria for spots was employed. As the reference intensity $I_R$ has to be determined with a large radius (toward 1 μm), an exclusion zone twice this radius had to be taken to search for any adjacent localization that would compromise the intensity estimation. As a consequence, many localizations given by the localization software were rejected by TRABI due to impaired estimation of $I_R$. To compensate, significantly longer image stacks had to be recorded to ensure structural consistency in the reconstructed images regarding the Nyquist-criterion. Though this was already an inconvenience for common SMLM organic fluorophores, which exhibit high repetition counts, it can be a significant obstacle for imaging approaches utilizing photo-activatable or -convertible fluorescent proteins. Furthermore, in cases of sub-optimal photo-switching rates that lead to high spot densities, the TRABI approach was computationally expensive or could even fail to produce an image due to a high rejection rate of fluorescent spots.

For a reasonable large TRABI radii $I_R$ converges for both spots in both planes, thus Eq. (1) can be effectively simplified to:

$$I_{Fit,1,2}(z) = \frac{I_{Fit,1} - I_{Fit,2}}{I_{Fit,1} + I_{Fit,2}} \tag{2}$$

As Eq. (2) demonstrates, in order to achieve three-dimensional BP imaging, there is no need for an actual photometric TRABI analysis employing two apertures anymore, since $I_R$ is canceled out. Therefore, dTRABI allowed for higher emitter densities, was quicker and required less input parameters. In the most straightforward way, the fit intensity of the localization software can be used.

*Calibration and allocation.* Raw calibration curves were generated by linearly moving the focal plane through the sample plane while imaging a single-molecule surface under dSTORM conditions as previously described[59]. For this, a surface of BSA was doped with BSA molecules, attached to a short DNA sequence, labeled with Cy5. Fitting the raw emission patterns by independent Gaussians with the same fixed FWHM yielded axially dependent single-molecule intensity curves. An axially precise raw calibration function γ was derived according to Eq. (2). The running median (binning width 25 nm) of the raw data was fitted with a high-order polynomial to generate the axial lookup table.

After an initial rough alignment between the channels, experimental localizations from both optical channels were assigned by a framewise, linear nearest neighbor analysis. Here, the distance threshold was set to 500 nm, which seemed a reasonable value for the robust allocation between channels in a semi-sparse single-molecule environment. From these sets of localizations, the axially dependent intensity quotient $I_{Fit,1,2}$ (Fig. 2b) was calculated and roughly allocated to the look-up table (LUT). The final axial coordinate was determined by a linear interpolation of $I_{Fit,1,2}$ between its "left" and "right" nearest neighbor coordinate of the LUT. Obtained axial coordinates were corrected for the refractive index mismatch as previously described[59].

*Drift and tilt correction.* Since the plasma membrane can be seen as flat over the whole-cell scale, we reckoned that it can be used as its own fiducial marker. We traced the spatio-temporal axial footprint of the entire membrane by fitting the raw localizations by a high-order polynomial in time. This is followed by the straightforward temporal linearization of the localization data, leading to a stable axial mean value over time. In order to instantly assess the quality of three-dimensional SMLM data we suggest looking for white regions in the color-coded image. In our experience, the abundance of these features usually suggests significant axial drift. However, even a subtle axial drift, which cannot be easily recognized by eye, will be detected by the approach described above. Additionally, non-linear drift, commonly occurring due to heating and resting of threads, can be accounted for.

Since we regularly observed a subtle axial tilt in the nanoscopic color-coded images due to the usage of a round coverslip in a magnetic holder, we developed a simple correction workflow (Fig. 3). The axial tilt of the sample is extrapolated by fitting a plane to the raw image data (pixel size 100 nm). Afterwards, the data was linearized by simply subtracting the precise local plane-value from the raw localization.

For all experimental data, axial drift correction was performed prior to a tilt test and correction. Exemplary drift and tilt data in Fig. 3 were simulated for better illustration, but based on real data (the displayed cell initially displayed both significant drift and tilt). Right columns represent experimentally corrected real data on which the artificial drift and tilt were projected.

*Localization precision calculation.* Drift and tilt-corrected localizations were tracked in time by determining the three-dimensional nearest neighbor distances of

localizations in consecutive frames. Localizations constructing a track with a total inter-localization distance of less than 75 nm were considered to stem from the same fluorophore. For each sample type, we combined the nearest-neighbor tracks from all recorded independent fields of view, calculated the deviation from the mean coordinate of each track for all relevant spatial coordinates and derived a normalized histogram (Supplementary Fig. S9). By fitting these distributions with a Gaussian, we derived the localization precisions as the standard deviation of the mean.

*TRABI and Fourier ring correlation (FRC) analysis.* Standard TRABI-biplane files were created with a TRABI radius of 8.5 camera pixel, a base-jump of 2 frames, an exclusion zone factor of 2, a highlander-filter of 250 frames and a number of averaged background frames of 5. Three-dimensional data sets were then sorted into groups of localizations apparent in even and odd numbered frames, respectively, which were rendered to high-resolved images in rapid*STORM* with 10 nm pixel size. FRC was performed on these two respective images, using the NanoJ-Squirrel Fiji plugin[72] with the input of 20 segments per dimension. Since the resulting FRC resolution maps showed a prevalence for extreme outliers, we chose the median FRC resolution over the entire displayed image as resolution criteria (Supplementary Fig. S10).

**Axial localization distribution analysis.** Localization files obtained from rapid-STORM were loaded and processed in Fiji[73] with custom written scripts[74]. First, all localization coordinates in $x$, $y$ and $z$ were used to generate a quantitative stack of 2D images with 20 nm pixel size using a separation of 20 nm in $z$ (i.e., $z$-stack). The numeric value in each pixel was equivalent to the number of localizations. Then the $z$-stack was segmented into defined ROIs. The interior of the cells was manually selected to serve as boundary ROI, in which the edges of the cells were spared. Within this master ROI a set of squared ROIs with $2 \times 2 \, \mu m^2$ area was automatically generated. The ROIs were allowed to be confined by the border of the master ROI and only kept when more than 75% of the ROI area, i.e., >3 μm were preserved (Supplementary Fig. S11).

Afterwards, the localization density of each ROI was analyzed by accumulating the gray values per ROI within the image stack, i.e., stepwise in $z$ every 20 nm, and plotted as a function of $z$. The plot was then fitted to a bi-Gaussian function of the form:

$$y = a_1 \exp\left(-\frac{1}{2}\left(\frac{x - m_1}{s_1}\right)^2\right) + a_2 \exp\left(-\frac{1}{2}\left(\frac{x - m_2}{s_2}\right)^2\right), \quad (3)$$

with $a_{1,2}$ as amplitude, $m_{1,2}$ as mean value and $s_{1,2}$ as standard deviation of each Gaussian.

To characterize the axial distribution of localizations of CD4-WT, CD4-CS1 and CD45, three parameters were derived from this fit.

(1) The z-distribution width (Fig. 5e–g) calculated according to:

$$z_w = \left(m_2 + \frac{FWHM2}{2}\right) - \left(m_1 - \frac{FWHM1}{2}\right) \quad (4)$$

for $m_2 > m_1$ with $FWHM_{1,2} = s_{1,2} \times 2.355$;

(2) the peak-to-peak distance (Supplementary Fig. S13) according to $top$-$p = |m_2 - m_1|$; and

(3) the width difference of the two Gaussians (Supplementary Fig. S13) according to:

$$\triangle_{FWHM} = |FWHM_1 - FWHM_2|. \quad (5)$$

**Quantifying axial receptor distribution.** For quantifying the obtained axial receptor distribution in Fig. 5f, we fitted the histogram of $z_w$ for CD4 WT, CD4 CS1 and CD45 with a model of Gaussian functions. CD4 WT was fitted using a mono-Gaussian and yielded the mean value $\mu$ and standard deviation $\sigma$, the range of $z_w$ in Table 1 was stated as $\mu \pm FWHM/2$, with FWHM as full width at half maximum of the distribution. CD4 CS1 and CD45 $z_w$ histograms were fitted using a bi-Gaussian yielding $\mu_1$, $\mu_2$ and $\sigma_1$, $\sigma_2$ as mean and standard deviation, respectively. The range of values in Table 1 was stated as $\mu \pm FWHM/2$. The p-p distributions (Supplementary Fig. S13) were modeled with a mono-exponential decay, and the inverse of the decay constant of the fit ($\tau$), i.e., the value at which the amplitude is reduced to 36.8%, was stated as mean value. The distribution of $\Delta_{FWHM}$ was modeled with a mono-Gaussian, thus obtaining $\mu$ and $\sigma$ as mean and standard deviation, respectively; the range of values in Table 1 was stated as $\mu \pm FWHM/2$.

**SOFI.** Sample preparation, coverslip coating and image acquisition were performed as described above for SMLM imaging. The raw data were analyzed using balanced SOFI algorithm as described before[28,87].

**Single-cell calcium measurements**
*Calcium sensor.* Plasmid DNA with the ultrafast, genetically encoded calcium sensor GCaMP6f$_u$ was a kind gift from Katalin Török (St George's University of London) and Silke Kerruth. The coding sequence was amplified using primers

AATAGATCTGCCACCATGGGCTGCGTGTGCTCCTC and AATGGATCCTC ACTTCGCTGTCATCATTTGTACAAA and subcloned into pXJ41 vector using *BglII* and *BamHI* restriction sites as described before[88].

*Sample preparation.* For calcium measurements, the coverslips were coated with 0.01% (w/v) PLL (Sigma-Aldrich) or 2 M glycine as described above. For coating with OKT3 (anti-CD3ε) antibody, clean coverslips were incubated with 0.01 μg/μl OKT3 in PBS for 30 min at 37 °C. After a brief wash with PBS, the coverslips were used within a day. Prior to acquisition, coated coverslips were mounted into a ChamLide holder (Live Cell Instruments), filled with 500 μl of color-free medium, placed on the microscope and focused to the focal plane. The measurements were performed 20 h after transfection of Jurkat cells with the calcium sensor. For image analysis, cells were washed with PBS, resuspended in color-free RPMI-1640 media (Sigma-Aldrich) supplemented with 2 mM $L$-glutamine, 10 mM HEPES, 1 mM $CaCl_2$ and 1 mM $MgCl_2$ and dropped onto the prepared coverslip while running the image acquisition.

*The microscope setup and image acquisition.* Live-cell calcium mobilization imaging was performed on a home-built TIRF microscope consisting of the IX73 frame (Olympus), UApo N, 100×1.49 Oil immersion TIRF objective (Olympus) and OptoSplit II image splitter (CAIRN Optics) mounted on the camera port. Samples were illuminated using 200 mW 488 nm laser (Sapphire, Coherent) in a TIRF mode and the intensity was regulated by acousto-optic tunable filter (AOTFnC-400.650-TN, AA Optoelectronics). Fluorescence emission was detected by an EMCCD camera (iXon ULTRA DU-897U, Andor) with EM gain set to 200. Images were taken in 500 ms intervals with the exposure time set to 50 ms.

*Data processing.* Calcium mobilization was quantified by analyzing mean fluorescence intensity (MFI) changes in cells landing on coated coverslips over the period of 10 min. The area occupied by a landing cell was manually selected (ROIs) from the frame with maximal size of the cell contact. Five representative MFI kinetics of the cells landing on PLL-, glycine- or OKT3-coated coverslips are plotted in Supplementary Fig. S7. The peak intensity ($I_{max}$) was determined by finding a maximum MFI value in a track. The early phase of the measurement (an increase in intensity) represents a combination of the rise of background fluorescence due to cell spreading on the optical surface and the increase in specific sensor fluorescence caused by increased calcium in the cytosol. It cannot be used to monitor calcium response in cells landing on diverse surfaces. Therefore, to determine the extent of calcium response, we calculated the decrease in intensity after the maximal response. We first determined MFI values 5 min ($I_{max+5 \, min}$) and 10 min ($I_{max+10 \, min}$) after the maximal response. To avoid the impact of the intensity fluctuations, we calculated $I_{max}$, $I_{max+5 \, min}$ and $I_{max+10 \, min}$ as an MFI average of ten frames (ROIs). The kinetics of fluorescence decay was then calculated as a ratio $I_{max}/I_{max+5 \, min}$ and $I_{max}/I_{max+10 \, min}$ Higher values indicate a more rapid decay of the fluorescence, which indicates more active response. Fiji/ImageJ software and basic spreadsheet functions of Microsoft Excel were used for the analysis[73].

**Large-scale calcium measurements**
*Sample preparation.* For calcium measurements, the coverslips were coated with 0.01% (w/v) PLL (Sigma-Aldrich) or 2 M glycine as described above. For coating with anti-TCR antibody, clean coverslips were incubated with 0.01 μg/μl C305 (Merck-Millipore) in PBS for 30 min at 37 °C. Glycine-coated coverslips were used immediately, PLL- and antibody-coated coated coverslips within a day after preparation.

Jurkat CD4-KO cells and primary CD4+ T cells were diluted to a final concentration of $1 \times 10^6$/ml in 1 ml of culture media with 1 μl of 1 mM Fluo-4 AM (Invitrogen) and 2.5 μM probenecid (Sigma-Aldrich) and incubated for 30 min in a humidified incubator at 37 °C, and 5% $CO_2$. Cells were then collected by centrifugation at room temperature for 5 min ($500 \times g$ for Jurkat CD4-KO cells and $1500 \times g$ for primary CD4+ T cells). Cells were resuspended in 500 μl of pre-warmed color-free RPMI-1640 medium (Fischer Scientific) supplemented with 2 mM L-glutamine, 10 mM HEPES, 1 mM $CaCl_2$ and 1 mM $MgCl_2$ and 2.5 μM of probenecid and immediately transferred to the coverslips via tubing connected to the ChamLide holder with image acquisition already running.

*The microscope setup and image acquisition.* Live-cell calcium influx measurement was performed on the IX73 frame (Olympus) equipped with ×20 objective (UPLXAPO20X, Olympus), LED epifluorescence illumination system (CoolLED pE-4000, CoolLED) and sCMOS camera (Zyla 4.2P-CL10, Andor). Coated coverslips were mounted into the ChamLide holder with micro-fluidic system attached to a live-cell environmental chamber (Uno, Okolab S.r.l.) set to 37 °C. Acquisition time for one frame was 250 ms with illumination using 470 nm set to 10%. Cells were imaged for 15 min (3600 frames) after injection into the holder. Afterwards, ionomycin (1 μM; Sigma-Aldrich) was added to the sample to measure maximal response for each cell. The acquisition was controlled by μManager software (ver. 1.4.24; ref. [89]).

*Data analysis.* Data were first converted to an 8-bit mode and background was subtracted using the method of rolling ball with a radius of 50 pixels using Fiji/ImageJ software (version 1.53f51; ref. [73]). Calcium response of each detected cell

was calculated using previously reported algorithm CalQuo2[90]. Cell fluorescence in each frame was normalized to the maximal fluorescence detected after stimulation with ionomycin. The first frame was individually set for each cell based on their asynchronous appearance in the field of view. To avoid the impact of signal fluctuation, fluorescence from 10 frames around the selected time point was averaged and plotted for each cell using Origin PRO 2021 software (OriginLab, ver. 9.8.0.200). In addition, the ratio of cells, in which no calcium mobilization after the contact with coated coverslips was observed and those, which exhibit detectable calcium response was calculated.

Three independent experiments were measured using Jurkat CD4-KO cells for each glass-coating protocol (0.01% PLL, glycine and activating antibody anti-human TCR (C305, Millipore)). Three independent experiments were measured using human primary CD4+ T cells and, in total, five independent measurements were recorded for each coating protocol.

**Characterization of the cell surface in contact with a coated coverslip**
*Microscope setup.* For imaging of living Jurkat cells forming contact with a coated coverslip, a home-build inverted microscope system (IX71, Olympus) equipped with 150 mW 488 nm and 150 mW 561 nm lasers (Sapphire, Coherent) was used. Fluorescence emission was detected by an EMCCD (iXon DU-897, Andor) camera. Two acousto-optic tunable filters (AOTFnC-400.650-TN, AA Optoelectronics) provided fast switching and synchronization of lasers with a camera. The 488 nm laser beam was focused onto the back focal plane of an objective (UApoN ×100, NA = 1.49, Olympus). TIRF illumination was achieved using a manual micrometer-scale tilting mirror mechanism in the excitation pathway (Thorlabs). The system was controlled using the μManager software (version 1.4.22; ref. [89]).

*Analysis of the cell surface in contact with a coated coverslip.* For T-cell-coverslip contact analysis, Jurkat T CD4-KO cells were transfected with pXJ41-CD4-WT-eGFP plasmid 24-h prior to the measurement. Cells were harvested and transferred into color-free RMPI medium (Thermo Scientific) supplemented with 2 mM L-glutamine (Lonza) pre-warmed to 37 °C. In parallel, cleaned coverslips were mounted into a ChamLide imaging holder equipped with tubing adapters (Live Cell Instruments), coated as described above and washed with ultrapure water. Immediately after coating fluid removal through the attached tubing, cells were injected into the ChamLide chamber attached to the microscope stage. The emitted fluorescence was recorded on an EMCCD camera using TIRF illumination for 20 min. Images were acquired in 1 s intervals with 50 ms exposure time. The camera EM gain was set to 100. The experiment was performed at 37 °C using the environmental chamber (Okolab) with controlled temperature. Acquired images were processed using standard functions (ROI Manager, Plot Profile) of Fiji/ImageJ software (version 1.52p; ref. [73]).

For signal homogeneity analysis, 20 frames (50 ms frame rate) were acquired in the TIRF mode 5 or 10 min after injection of cells into the imaging chamber. EM gain of the camera was set to 300. The frames were summed to generate a collate image for further quantitative analysis using Fiji/ImageJ software (version 1.52p; ref. [73]). The data were further analyzed as described in the "Homogeneity analysis" section.

To evaluate the impact of fixation on cell morphology, Jurkat CD4-KO cells transfected with pXJ41-CD4-WT-eGFP plasmid were seeded on glycine-coated coverslips. First set of cells was imaged as described above for signal homogeneity analysis. Second set of cells was fixed 5 min or 10 min after seeding on coated coverslips. The fixation was performed as described for SMLM without immunostaining and post-fixation steps. The image acquisition and data processing were performed as described above for live-cell contact analysis.

Contacts of Raji, RAW264.7 and COS-7 cells with a coated coverslip were characterized after transfection with pXJ41-CD4-WT-eGFP plasmid 24-h prior to the measurement (see section "Cell culture and transfection of Methods"). The contact sites were monitored as described above for transfected Jurkat CD4-KO cells with two modifications. First, HILO illumination was used instead of TIRF. Second, two to five focal planes were imaged to visualize the pattern at the cell contact with a coated coverslip, but also of the cell body positioned >500 nm above the optical glass. Data processing was performed as described above for live-cell contact analysis.

*Interference reflection microscopy (IRM) of T-cell contacts with coated coverslips.* Interference microscopy was performed at 37 °C using Olympus FluoView1000 MPE confocal scanning microscope (Olympus) equipped with 488 nm laser (Sapphire, Coherent), ×60 water immersion objective (UPLSAPO, NA = 1.2, Olympus) and the environmental chamber (OKO Lab). Scanning laser light reflected from the glass-water interface was collected on a PMT detector without optical filter application with scanning speed 8 μs/pixel and Kalman denoising filter (4 lines mode). The acquisition was performed 5 min or 10 min after seeding of transfected Jurkat CD4-KO cells on PLL- or glycine-coated coverslips (as described above). Images were processed using Fiji/ImageJ software (version 1.52p; ref. [73]).

*Homogeneity analysis.* Fluorescence homogeneity (according to ref. [91]) was calculated with a custom written Python script. In short, raw images were converted to an 8-bit gray value pixel-matrix and an objective intensity threshold via the ISODATA method was applied. All gray values below the threshold were set to 0. Afterwards, the gray level co-occurrence matrix (*glcm*) was set up by comparing the gray values of pixels with a set offset (in our case 1) and angles (we chose the values

0, pi/4, pi/2 and ¾ x pi—corresponding to the horizontal, vertical and both diagonal directions). This results in four square matrices with their length equaling the number of gray levels in the picture (in our case 256 for 8-bit images) and the amount of occurring pairs, according to the indices, as entries. To analyze only the area of interest in the image, i.e., the cell, all entries of the *glcm* corresponding to the gray value 0 (i.e., the first row and column) were set to 0. Next the *glcms* were normalized, each entry was divided by the total number of co-occurrences so the sum over the whole matrix is 1. Afterwards the entries were weighed, according to their position in the matrix (entries were divided by $1 + (i, j)$, $i$ and $j$ being the indices respectively) and summed over to result in the homogeneity value for a particular offset and angle. Since we chose one offset and four angles, we averaged over the four homogeneity values to get the final value for the image.

**Cell viability assay.** To determine the viability of Jurkat cells immobilized on coated coverslips, a home-build inverted microscope system (IX71, Olympus) equipped with 150 mW 488 nm and 150 mW 561 nm lasers (Sapphire, Coherent) and the objective (UPlanSApo ×10, NA = 0.4, Olympus) was used. Fluorescence emission was detected by a sCMOS camera (Zyla-4.2-CL10, Andor). Two acousto-optic tunable filters (AOTFnC-400.650-TN, AA Optoelectronics) provided fast switching and synchronization of lasers with a camera. The system was controlled using the μManager software (version 1.4.22; ref. [89]).

Jurkat cells were transfected with mTurquoise-Farnesyl-5 plasmid (#55551, Addgene) 24 h prior to the assay. This was done to mimic the conditions used for all other experiments in this work but to avoid interference of the protein fluorescence with the dyes used in the assay. For the measurement, cells were washed with PBS, transferred into color-free RPMI (Thermo Scientific) medium supplemented with 2 mM L-glutamine (Lonza), 10% fetal bovine serum (Thermo Scientific), 10 mM HEPES and Non-essential amino-acid mixture (Thermo Scientific) containing 25 μg/ml (w/v) calcein-AM dye and incubated for 10 min at 37 °C in the $CO_2$ incubator. Cells were then transferred into a fresh, color-free RPMI medium supplemented with 2 mM L-glutamine and 7AAD viability stain (7-amino-actinomycin D; 1:20 dilution; eBioscience) and incubated for another 5 min at 37 °C in the $CO_2$ incubator. After loading with the dyes, cells were injected onto the coated coverslips in ChamLide chamber using attached tubing. Imaging was performed for 30 min (with 10-min acquisition intervals) from the time of the first cell-coverslip contact detection. Two randomly selected ROIs were selected for each time point to avoid the effect of the fluorophore photo-destruction. Calcein and 7AAD dyes were excited separately by 488 and 561 nm lasers, respectively, using defocused light to achieve homogenous illumination. Cells were also imaged under transmitted light to control their morphology. To avoid the impact of fluorescence fluctuations, five frames were collected for each channel, ROI and time point. The frames were then summed to generate a collate image for further quantitative analysis. Data analysis was performed in Fiji/ImageJ (version 1.52p, ref. [73]). Cells were counted by determining local maxima for each channel. Cell viability was calculated as a percentage of 7AAD positive (dying) cells in all detected cells (calcein positive). The experiment was performed at 37 °C using the environmental chamber (OKO lab) with controlled temperature.

**Characterization of cell mobility on coated coverslips.** Approximately $5 \times 10^5$ cells were resuspended in pre-warmed, color-free RPMI medium and seeded immediately on PLL-, glycine- or serum-coated coverslips mounted in pre-warmed Chamlide imaging chambers. Image acquisition was started before seeding cells on coverslips. Imaging was performed at 37 °C using a home-build microscope (IX81, Olympus) equipped with 20x phase contrast objective (LCAchN 20xPH, Olympus), a halogen lamp with the appropriate phase contrast aperture and the environmental chamber (OKO lab). Camera frame rate was 50 ms and EM gain to 50. The acquisition was stopped 15 min after the first contact of cells with a coverslip. Images were processed and movies created using Fiji/ImageJ (version 1.52p, ref. [73]).

**Confocal microscopy of CD4.** Confocal microscopy of Jurkat cells transfected with pXJ41-CD4WT-GFP or pXJ41-CD4CS1-GFP plasmids was performed as described before[92].

**Atomic force microscopy.** AFM topography images were collected with Dimension Icon AFM (Bruker Instruments). All images were measured in the Peak Force Tapping mode for fluids, using a probe holder for fluid operation and Scanasyst-Fluid probes (Bruker) with a tip radius of 20 nm and a spring constant of 0.7 N/m.

In total, 12 mm diameter coverslips were coated with PLL or glycine as described above. A bare coverslip washed with deionized water was prepared as a reference. The AFM probe was submerged in the solution in order to land on the coverslip surface. Due to a delicate nature of the samples and to avoid long measurement times, setpoint and number of lines were set to 400 pN and 256 × 256 lines respectively, with a scan rate of 1 Hz. Images with scan sizes of 1, 2 and 10 μm were collected.

**Statistics and reproducibility.** Statistical significance between the groups (surface coatings) in Fig. 1e, f was evaluated using one-way ANOVA test, in Fig. 1g using unpaired *t*-test with unequal variances (Satterthwaite's approximation). Boxplots show the median (middle bar), 25th and 75th percentiles (upper and lower limits of

the box), and whiskers extend to the most extreme data. Each data point is represented by a symbol, and median by a solid line as described in figure legends. The sample sizes are indicated in the main text and figure legends.

**Reporting summary**. Further information on research design is available in the Nature Research Reporting Summary linked to this article.

## Data availability

All source data generated and/or analyzed during this study are included in this article (and its Supplementary Information and Supplementary Data) or are available from the corresponding authors on reasonable request.

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

## Acknowledgements

We would like to thank Lilliana Barbieri (University of Oxford), Peter Kapusta, Silke Kerruth and Harsha Mavila for technical assistance and professional advice. We are grateful to Markus Sauer (University of Würzburg) for initial support of the project. C.F. would like to thank Laure Plantard and Jan Peychl from the MPI-CBG LMF for technical support. M.C. acknowledges funding from Czech Science Foundation (19-0704S), S.v.d.L. and L.H. acknowledge funding from Academy of Medical Sciences/the British Heart Foundation/the Government Department of Business, Energy and Industrial Strategy/the Wellcome Trust Springboard Award (SBF003\1163) and from the EPSRC through a doctoral fellowship (EPSRC studentship 2031229 and EPSRC Doctoral Training Partnership (DTP) Grant EP/N509760/1). The measurements at the Imaging Methods Core Facility in BIOCEV, Vestec, Czech Republic were supported by MEYS CR grant Z.02.1.01/0.0/0.0/16_013/0001775.

## Author contributions

C.F., S.v.d.L., and M.C. conceived the study. C.F., T.C., Z.K., D.G., D.A.H., and H.M. performed the experiments, C.F., T.C., D.G., G.J.G., L.H., S.v.d.L., and M.C. analyzed the data. A.R. measured and analyzed the AFM data. O.F. and T.B. provided research advice. C.F., S.v.d.L., and M.C. wrote the paper. All authors reviewed and approved the manuscript.

## Competing interests

The authors declare no competing interests.
