## [Peer Review File · Communications Biology]

Reviewers' comments:

Reviewer #1 (Remarks to the Author):

This is an interesting paper where the authors present a new approach to improve imaging, at single molecule resolution, of surface receptors. To do so they suggest an improved methodology that allows mapping of receptors, such as CD4 and CD45, at the nano-topography of the membrane, that is, with three-dimensional resolution information. This is particularly relevant given the newly appreciated role that lymphocyte membrane topography plays in cell signaling and receptor distribution. The new methodology can be divided in two main principles. The first is the development of an experimental setup that allows imaging of an untouched, unstimulated T cell where all fine structures are preserved, as an attempt to gain access to the elusive "resting lymphocyte" surface. The second is to then gather information using a bespoke imaging setup on the distribution of important membrane receptors. The authors explore using glycine as an inert coating agent to trap T cells coupled to the optimisation of the previously described TRABI method. The usage of glycine seems particularly interesting and warrants careful investigation as an alternative to other coatings such as PLL which have been shown to stimulate lymphocytes. Overall, the approach here described is timely and relevant as it could allow researchers to gain a better insight into unperturbed cell surface. This approach could become a Holy Grail for molecular immunologists (and others) as the field is clearly longing for the perfect methodology that allows artefact free analysis of the structures and the receptor distribution in untouched T cells. However, the manuscript could be greatly improved if additional details and data are included. While presenting a very promising new pipeline for single molecule imaging, the biological significance of the findings is preliminary and should be strengthened to better address general but key concerns in the field. Overall the manuscript is clearly written, and figures are well presented.

1. Glycine is here presented as a potential alternative to allow imaging of unperturbed lymphocytes. The authors decide to validate this approach by looking at calcium responses and protein distribution upon T cell interaction with glycine coated surfaces.

1.1 Prior to looking at surface receptor distribution, it seems that it would be important to image the actual cell membrane using this approach (glycine + dTRABI combo). Attempts to accurately describe the nanotopography of cell surface receptors must take into account membrane heterogeneities. Imaging of a membrane dye could offer very significant information in this regard (on Fig. S11 the authors acknowledge the potential different morphologies of the cell after landing on the glycine coated plate).

1.2 Related to the previous point, in Fig. 1c, could the heterogeneity of CD4 (and CD45) be due to membrane ruffles and/or dynamics?

1.3 It would be helpful to know what is the percentage of cells that exhibit responding v non-responding behaviour (calcium release). This data is missing but it is extremely important to establish glycine as a valid alternative to other coatings. Overall very few cells were used and the authors present one representative calcium release trace for each condition, it would be better to show more traces given the novelty of the approach. Moreover, the glycine induced T cell response will dictate how confident one can be during imaging acquisition on pre-fixed cells (where the activation status of the cells is lost).

1.4 Importantly, the authors chose to show data in Fig 1e for intensity of $I_{\max}+5$ min or 10 min relative to to I_{\max} . Have the authors considered using the relative intensity before the I_{\max} ? Wouldn't this be closer to ascertain where there was a calcium release in the first place? As is, the data shown in Fig. 1e seems to be related to the transientness of the calcium release and says little about the actual calcium response upon cell contact with the surface.

1.5 Also, regarding glycine, the authors state that "We were thus unable to determine the exact thickness and stiffness of the glycine layer", could this per se influence the cell topography as cells penetrate through the glycine? How uniform is the glycine coating distribution? Are there "glycine-free" areas?

1.6 Can the authors discuss whether the heterogeneity that is shown in figure S2 regarding the glycine coating (using AFM) may contribute to imaging artefacts? Is there any way to account for this when doing SMLM?

2. Introducing fiducial markers for optimal axial localization over an extended imaging time has

proven to be extremely important to accurately describe the data, particularly given the membrane heterogeneity and, for example, sample drifting, which as the authors know presents a serious technical challenge, specially in SMLM data. The method chosen by the authors is rather interesting but how does it compare to other more commonly used fiducial markers such as beads?

3. The dTRABI approach seems a very promising alternative way of acquiring 3D information.

3.1 When the authors irradiate the whole cell in epifluorescence could it lead to preferential bleaching of certain structures over others, has this been accounted for, if so, how?

3.2 Given that different molecules are expressed at different densities and that the "resolution-enhancement is more pronounced for structures with a medium to high local density" is this easily solved with the quantitative localization analysis, how was this accounted for?

4. When staining CD4 and CD45 the authors used whole antibodies. Can the authors confirm whether antibody was used on live or fixed cells? This was unclear from the methods section. Furthermore, antibodies can introduce biases in the labelled population due to fixation, epitope, affinity and/or protein density. The authors mention that a CD4 labelled with a fluorescent protein (FP) was used for sorting etc. Can the authors compare the distribution of antibody labelled CD4 with CD4-FP? This is an important point to review and clarify.

5. In figure 7 where the authors suggest that CD45 might be excluded from CD4 enriched areas all that is presented is 2D SMLM data, do the authors possess now any more information from multi-colour dTRABI analysis? This would be key for the claims in Fig. 7c.

6. It's a bit confusing throughout the paper exactly which reagents were used, i.e, for CD4 there is data with CD4-GFP, CD4-mEos and direct label with a full antibody against CD4, the manuscript would benefit from a clearer description of the experiments and reagents used.

7. In the methods section, Jurkat T cells (clone E6-1 and not E6)

8. In the Jurkat CD4-KO did the authors tested the potential downregulation of Lck, as this could influence the activity of these cells.

9. Did the authors try any other fixation method? Several reports have claimed that inadequate cell fixation methods could lead to artifactual molecular redistribution by the antibodies themselves. Do the authors have any indication that fixation conditions do not affect the outcome? For example, PFA (paraformaldehyde) vs PFA+GA (glutaraldehyde)?

Reviewer #2 (Remarks to the Author):

The manuscript by Franke et al. presents an improved SMLM method with axial resolution capable of resolving nanotopography of the basal membrane of adherent cells, and apply it to investigate the organisation of two key proteins involved in activation of T lymphocytes, CD4 and CD45. They improved and simplified their previous SR imaging approach (TRABI), which will help its wider adoption. They also employed an alternative protocol for cell immobilisation, necessary for SMLM, using glycine instead of PLL coating to reduce non-specific activation of T cells and prolong their viability – a welcome step towards controlling experimental conditions in this sensitive application. They finally aimed at elucidating the origin of CD4 clusters in unstimulated T cells.

The manuscript is very well written, clearly structured, and graphical material nicely presented. The methodological contributions of the paper, both regarding imaging modalities and sample preparation, are original and of interest to the wider field. Regarding the specific findings about the organisation of the investigated protein clusters and their biological role, though, I would encourage some further corroboration.

My main questions and suggestions are the following:

1. In several places the authors claim that "glycine-coated optical surfaces ... preserve the native membrane morphology of immobilized cells". Without quantification and a proper experimental control (e.g. with cells suspended in a hydrogel, comparison of basal vs apical membrane, or temporal evolution from very early after landing as they attempted qualitatively), though, this should be toned down to comparison with PLL (e.g. "cells flatten less than on PLL").
2. If there is any other way of characterising the mechanical properties of the glycine layer, that would be very helpful to the community.
3. The authors thoroughly analyse the distributions of axial localisations, approximating them with bi-Gaussian functions and comparing the extracted parameters. However, I find some of the descriptors questionably informative, especially when the second component is small or largely overlapping with the first (Fig S6a). Would it perhaps help to fix one component (main peak) and then fit only the shoulder, which varies most between the samples? Or perhaps complement z_w with some interquartile distance?
4. The reader remains hungry for further insights into the meaning of the devised proteins' axial distributions, e.g. on the origin of the two populations (localisations on separate structures/folds of the membrane?). Please discuss also to what degree this will be relevant to a lymphocyte interacting with another cell with a less flat surface.
5. Central regions of the cells in dTRABI images of CD45 and CD4 look very "flat". Please discuss the potential effects of the localisation filtering on biasing the results. It would be instructive to compare raw total intensities to those generated from the reconstructed images after filtering, to see if any "features" are systematically filtered out (by intensity, z , etc).
6. From the presented dTRABI images of CD4 CS1 cells (Fig S6 & S9), it seems like 3 out of 5 are very similar to the WT. A comparison of cell-by-cell distributions (instead of ROI-wise) would therefore also be informative. Can this mutation influence the attachment/spreading of the cell?
7. Can the observed exclusion of CD45 from the CD4-rich zones be explained also by steric effects at the contact zone with the surface, not (solely) from different distributions along the protrusions? To support authors' interpretation, in my view, the case in Fig 7d-e (cell just landing) is much more compelling than the case in Fig 7a-b (fully spread cell) where distribution of CD4 seems very different from other images with dTRABI.
8. In the Discussion, please compare dTRABI to other SMLM (and other SR) approaches with sub-diffraction resolution (nicely listed in the Introduction), to help readers decide which one suits their application best. Please also discuss potential limitations and biases of the SMLM approach, requiring long acquisition and immobilisation, e.g. detecting stationary vs mobile protein clusters, potency to induce clusters by immobilisation, etc.
9. Abstract: "near isotropic 3D nanometer resolution" – please specify (e.g. "resolution around A/B in XY/Z on such and such samples")
10. Fig 1e: please describe in the caption whether each square represents a single cell or a repeated experiment (with N cells)
11. Fig 3: the corrected images in panel a and b are the same, which is illustrative, but confusingly synthetic (apparent need for just one OR the other correction on the same dataset). Please clarify,

e.g. indicate the order of corrections in a real workflow.

12. Fig 5: where is the $z = 0$ plane in the distributions?

13. Fig 5: are panels h-i missing or is the caption redundant?

14. Fig 6: what exactly does "lacking CD45" refer to?

15. Fig S5: Do you report lateral or axial resolution on these 3D data sets?

16. Page 22, T cell viability assay, line 13: "fist" -> "first"

I thank the authors for this interesting and relevant work. When they revise the manuscript, I will be happy to review it again.

Iztok Urbancic

Reviewer #3 (Remarks to the Author):

In their current ms. Franke et al. focus on the localization of membrane proteins in three dimensions below the diffraction limit of visible light. More specifically, they showcase two approaches to reach that goal: The first deals with glycine coating of glass surfaces as a means to provide a substrate for cell attachment without inducing substantial changes in cellular morphology (i.e. loss of microvilli and other membrane protrusions). The second concerns improving existing single molecule localization microscopy (SMLM) methodologies for the third dimension with the use of highly inclined and laminated optical sheet illumination (HILO). The motivation here is to reduce cellular background by exciting predominantly cellular membranes in contact with the glass-surface preparation (a requirement for single molecule fluorescence detection) yet not limiting fluorescence to 100-200 nm into the specimen, as would be the case sample excited via evanescent light resulting from Total Internal Reflection modalities. To make their case, the authors image CD45 and CD4 expressed on Jurkat T-cells. They provide data suggesting differential membrane localization with regard to membrane protrusions: While CD4 but not its palmitoylation-deficient appear closer to the glass surface, CD45 seems at least in part be further away from the glass plane. Given the undeniably important implications of membrane topology and adequate quantitation for understanding cellular function I am highly in favor of exploring new ways and enhancing already existing methods to conduct 3D superresolution microscopy. Yet at the current stage I consider the study not sufficiently compelling to warrant publication in Communications Biology. Reasons for my overall assessment are stated below:

1. Overall presentation. I find both the introduction and the discussion section redundant and not written in a concise fashion. A lot of text deals with issues that have no direct relevance for the data shown. What exactly is the narrative? Is it biology or methodology? If it is the first, the message is too limited yet the methodological part receives too much attention. If it is the latter, then why spend much effort dealing with potential (yet not proven) biological implications? Instead, showing more evidence for method robustness and integrity would help. Figures 2, 3 and 4 could be combined to one figure. The same is true for figures 5 to 7.

There are statements I find close to overstating.

I am moreover of the opinion that in modern (and space restriction-free) days of online publishing methodologies should allow for easy data reproduction without the need for consulting referenced papers. This is clearly not the case here.

Furthermore, it took me a while to find out the specifics of the staining / visualization procedures employed for CD4 (antibody or genetic GFP fusions). More clarity in the results and methods section would help.

2. Glycine-coating. This is an interesting aspect of the ms., which would in my view clearly benefit from providing answers to a number of questions: How mobile are Jurkat T-cells on such surfaces? Do they crawl and scan their environment or do they stay put? Do primary T-cells attach to such surfaces? I understand the value of imaging T cells in a resting state (prior to antigen recognition),

but what would this form of attachment represent in the real world (resting T cells reside in different microenvironments such as skin, other organs, secondary lymphatics, blood, endothelial cells etc...). I feel categorizing glycine attachment as the most physiological for the preservation of filopodia is subjective and not necessarily true. Regardless, such filopodia are inherently interesting and deserve investigation, and here glycine coating may make the difference. But clarification in the text would position the use of glycine in a more adequate context. How about other cell types? Dendritic cells, B cells, macrophages? Do they attach? A methodology paper should address this and a simple experiment would tell immediately.

3. Choice of T-cells: Jurkat T-cells. I surely understand the practical benefits of using Jurkat T-cells, and obviously, to make a methodological point, the use of Jurkats may suffice. However, Jurkats are far removed from primary T-cells with regard to their physiology, and it shows: Primary T-cells do not undergo apoptosis when in contact with poly-L-lysine coated surfaces, nor do they flux calcium. The same is true for the use of protein-functionalized planar glass-supported lipid bilayers. Primary T-cells require antigen for activation without exception (while Jurkats flux spuriously). The operational need for Jurkats has vanished progressively with lenti-/retroviral expression systems and the emergence of CRISPR-CAS9. I hence find their use irrelevant if not at times misleading for drawing biological conclusions.

4. CD4 mutant. What is the evidence that the palmitoylation-deficient mutant is effectively transported to and predominantly present on the cell surface? For immunofluorescence on fixed but impermeable membranes this is not an issue, yet for any approach involving XFP(mEos/psCFP2 etc.)-gene fusions this may become an obstacle, in particular in view the HILO illumination scheme. An alternative explanation for locating this mutant further away from the glass surface may indeed be its association with internal membrane compartments. For the palmitoylation-deficient mutant of the Linker of Activation of T cells (LAT) this is certainly the case and has led to wrong conclusions in the past. The paper cited by the authors to support their claim that CD4 is clustered (while the mutant is not) is based on the use of CD4-fusion proteins.

5. SMLM. Excessive blinking (for PALM and dSTORM) as well as partial transfer into a reversible dark state (dSTORM) renders SMLM vulnerable to cluster artefacts. In part depending on their chemical nature a few individual fluorophores give rise to unpredictable blinking which cannot be accounted for by simple merging algorithms. To make things more complicated, only a small subset of originally available fluorophores may indeed be available for dSTORM (following irreversible ablation of the majority of fluorophores) producing an exaggerated pointillistic image with additional cluster artefacts. This needs to be addressed. A simple way could be to take an image of the cell surface prior to bleaching/photoswitching and compare this image with a SMLM image which has been computationally convolved / blurred using the FWHM employed for localization (i.e. reversing the SMLM process). If the SMLM image is correct, the intensity distribution of the convolved image matches that of the first ensemble image taken.

6. Main biological claim of the ms. that CD4 and CD45 segregate to different sites on microvilli. The evidence for lack of colocalization (shown in figure 7; a larger representation would help) fails to convince me. If this were the case, microvilli displayed in X-Y would clearly show this behavior, but they do not. The tips are never green (i.e. CD4 positive but CD45 negative) but always white (positive for both). Also, if conclusive the single color analysis provided in figure 4 would rather contradict such a scenario. However, the results shown in figure S9 let me question the validity of comparisons which are based on single color detections, as there appears to be considerable variability among different cell contacts. This leads me to conclude that dual-color colocalization is the only reliable means to derive topographical differences.

7. Validity of the 3D-nanotopography. Evidence that is independent of the showcased method would be helpful and compelling and should be in reach. Because the large majority of the 3D-structures observed in the SMLM images are large enough to be resolvable above the diffraction limit, a convincing case could be made by comparing the 3D superresolution plots with diffraction-limited RICM images taken from the same contacts. Both methods are sensitive within similar Z-ranges.

Reviewer #1 (Remarks to the Author):

This is an interesting paper where the authors present a new approach to improve imaging, at single molecule resolution, of surface receptors. To do so they suggest an improved methodology that allows mapping of receptors, such as CD4 and CD45, at the nano-topography of the membrane, that is, with three-dimensional resolution information. This is particularly relevant given the newly appreciated role that lymphocyte membrane topography plays in cell signaling and receptor distribution. The new methodology can be divided in two main principles. The first is the development of an experimental setup that allows imaging of an untouched, unstimulated T cell where all fine structures are preserved, as an attempt to gain access to the elusive “resting lymphocyte” surface. The second is to then gather information using a bespoke imaging setup on the distribution of important membrane receptors. The authors explore using glycine as an inert coating agent to trap T cells coupled to the optimisation of the previously described TRABI method. The usage of glycine seems particularly interesting and warrants careful investigation as an alternative to other coatings such as PLL which have been shown to stimulate lymphocytes. Overall, the approach here described is timely and relevant as it could allow researchers to gain a better insight into unperturbed cell surface. This approach could become a Holy Grail for molecular immunologists (and others) as the field is clearly longing for the perfect methodology that allows artefact free analysis of the structures and the receptor distribution in untouched T cells. However, the manuscript could be greatly improved if additional details and data are included. While presenting a very promising new pipeline for single molecule imaging, the biological significance of the findings is preliminary and should be strengthened to better address general but key concerns in the field. Overall the manuscript is clearly written, and figures are well presented.

Authors: We are very thankful for the positive evaluation of our work and the provision of several critical comments, which, we believe, led to an improved manuscript.

Reviewer #1: 1. Glycine is here presented as a potential alternative to allow imaging of unperturbed lymphocytes. The authors decide to validate this approach by looking at calcium responses and protein distribution upon T cell interaction with glycine coated surfaces.

1.1 Prior to looking at surface receptor distribution, it seems that it would be important to image the actual cell membrane using this approach (glycine + dTRABI combo). Attempts to accurately describe the nanotopography of cell surface receptors must take into account membrane heterogeneities. Imaging of a membrane dye could offer very significant information in this regard (on Fig. S11 the authors acknowledge the potential different morphologies of the cell after landing on the glycine coated plate).

Authors: The reviewer is right that it would be ideal to map membrane morphology first and then to overlay receptor distribution on such 3D map. Unfortunately, non-selective membrane dyes, evenly distributing over the plasma membrane are, to the best of our knowledge, still unavailable. Some membrane dyes (e.g., FM1-43 and its fixation-competent variant) were regularly used for membrane staining in confocal microscopy. However, such probes exhibit preferential localisation to different areas of the membrane, according to their preference for lipid bilayers with specific biophysical properties. This is regularly observed for membrane dyes in synthetic model membranes (Cebecauer and Šachl 2016). Moreover, and this is probably the most limiting factor, such dyes have poor photophysical properties, which exclude these fluorophores from the use in

SMLM - especially 3D methods requiring high photon yields. The dyes with better photophysical properties very poorly label plasma membrane due to their rapid redistribution to the intracellular membranes (e.g., Dil, DiD, MemBright variants – at least in T cell lines tested in our laboratory).

However, to partially answer this question, we can deduce cell surface morphology by combining the data on CD4-CS variant from our previously published work (2D SOFI; (Lukes, Glatzova et al. 2017)) with the distributions presented in this work (3D dTRABI). CD4-CS variants exhibits random distribution on T-cell surface and, thus, should indicate complex morphology in most of imaged cells (Fig. 5c and Supplementary Fig. S13).

1.2 Related to the previous point, in Fig. 1c, could the heterogeneity of CD4 (and CD45) be due to membrane ruffles and/or dynamics?

Authors: Membrane ruffles can certainly contribute to the distribution of surface receptors. Indeed, published SEM images demonstrate various protrusion types at the surface Jurkat T cells, including ruffle-like structures (Kim, Mun et al. 2018). Ruffles were also shown to influence receptor dynamics on immune cells (Welliver, Chang et al. 2011). Contribution of membrane protrusions other than microvilli to receptor distribution is now referred to in the revised manuscript.

1.3 It would be helpful to know what is the percentage of cells that exhibit responding v non-responding behaviour (calcium release). This data is missing but it is extremely important to establish glycine as a valid alternative to other coatings. Overall very few cells were used and the authors present one representative calcium release trace for each condition, it would be better to show more traces given the novelty of the approach. Moreover, the glycine induced T cell response will dictate how confident one can be during imaging acquisition on pre-fixed cells (where the activation status of the cells is lost).

Authors: In our work, we tested non-specific stimulation of T cells by the optical surface coated with glycine or PLL (and stimulating antibody as a control). Therefore, we could not use flow cytometry, which enables rapid analysis of thousands of cells. We used an imaging approach and a genetically encoded, membrane-associated calcium probe to determine the rapid calcium response at the plasma membrane of T cells in contact with the surface during the landing process. The relatively low number of analysed cells (compared to flow cytometry or low-resolution imaging of a cell population) is caused by the fact that we had to acquire the maximal response for each cell early after the first contact with the surface. Thus, we had to select only those cells, which we managed to follow from their first contact. We imaged individual cells using single-cell TIRF imaging. Only in one case, we were able to analyse two cells in the same sample. Incidentally, these two cells simultaneously landed at the optical surface. Due to a low number of measured cells, we cannot clearly state what proportion of cells did/did not respond to the stimulus. Our data indicate that all cells responded to the antigen-like stimulus (anti-TCR antibody). However, non-responsive cells probably did not spread on the surface and, thus, were not used for further analysis in our experiments.

We would like to thank the reviewer for the suggestion to present more tracks of calcium mobilisation per each condition. We agree that presenting more representative tracks will better highlight the difference between cells immobilised on glycine- vs PLL-coated glass coverslips. Five tracks are presented in the Supplementary Fig. S7 of the revised manuscript. We did not add more tracks to avoid overcrowding of the graph.

1.4 Importantly, the authors chose to show data in Fig 1e for intensity of $I_{\max+5}$ min or 10 min relative to I_{\max} . Have the authors considered using the relative intensity before the I_{\max} ? Wouldn't this be closer to ascertain where there was a calcium release in the first place? As is, the data shown in Fig. 1e seems to be related to the transientness of the calcium release and says little about the actual calcium response upon cell contact with the surface.

Authors: We agree with the reviewer that increase of the signal after stimulation compared to the baseline (non-stimulated cells) is the standard way for determination of calcium response. True, the values presented in Fig. 1f resemble transientness of the calcium release. The text of the corresponding figure caption (Fig.1) was revised accordingly. However, using an imaging-based approach to determine calcium response of a cell stimulated by different surfaces, we had to image cells from the first contact with the surface and follow the changes in cell fluorescence with time. We could not measure the baseline fluorescence. Our TIRF imaging approach does not enable visualisation of cells before the first contact with the surface. The increase in fluorescence signal during the early phase of cell 'stimulation' could not be used for determination of cell calcium response due the contribution of two processes to fluorescence signal during this phase: i) spreading of a cell on the surface (the cell has a background fluorescence and its contribution to the detected signal increases with cell spreading), and ii) changes in the fluorescence intensity of the probe due to increased intracellular calcium (the signal we are interested in). Since we are not aware of any well-established, 'inert' surface, on which our cells could spread in the absence of unspecific stimulation, we could not determine how much cell spreading contributes to the signal increase. Therefore, we decided to use signal decrease to compare cell stimulation efficiency by different surfaces. To avoid the impact of signal fluctuations, we calculated two values for each cell ($I_{\max+5}$ and $I_{\max+10}$). Jurkat cells are well known for the transientness of calcium levels after antibody stimulation. We thus believe that these values, together with presented signal traces, well document the superiority of glycine coating over PLL to avoid non-specific stimulation of cells.

1.5 Also, regarding glycine, the authors state that "We were thus unable to determine the exact thickness and stiffness of the glycine layer", could this per se influence the cell topography as cells penetrate through the glycine? How uniform is the glycine coating distribution? Are there "glycine-free" areas?

Authors: Suspension cells cannot be fixed on uncoated coverslips. This would lead to their washing off the coverslips. Therefore, if there are any areas without glycine coating, we would not be able to image cells, which incidentally landed in these areas.

A glycine-coated surface is probably heterogenous in terms of its height, however, due to the softness of this material, we were unable to measure its topography. Such softness of a glycine coat also suggests that it does not affect T-cell surface morphology. On the other hand, we observed disappearance of some protrusions on T-cell surface on glycine-coated coverslips after a longer incubation (> 15 min). This is probably caused by the fact that glycine forms only a very soft and narrow layer on the glass and the T-cell surface is still affected by g-force and the presence of hard matter (glass), which is not that common in biological tissues. Therefore, we speak about improved preservation of T-cell surface morphology compared to PLL. Glycine-coating expands the time window for measurements with a low impact of the optical surface on cell surface morphology.

1.6 Can the authors discuss whether the heterogeneity that is shown in figure S2 regarding the glycine coating (using AFM) may contribute to imaging artefacts? Is there any way to account for this when doing SMLM?

Authors: These heterogeneities in a glycine layer are caused by air bubbles and are smaller than any known T-cell surface structures (>40 nm). We do not expect their impact on T-cell spreading and surface protrusions.

2. Introducing fiducial markers for optimal axial localization over an extended imaging time has proven to be extremely important to accurately describe the data, particularly given the membrane heterogeneity and, for example, sample drifting, which as the authors know presents a serious technical challenge, specially in SMLM data. The method chosen by the authors is rather interesting but how does it compare to other more commonly used fiducial markers such as beads?

Authors: We agree with the reviewer, that for high-quality quantitative 3D imaging, axial stability is of utmost importance. We consciously refrained from using any fiducial markers for axial drift correction as the emitters usually add (out-of-focus) signal to the data set and are possibly not homogeneously distributed, which is especially critical in our case: There are two common choices on how to seed fiducials: 1. Seeding on the surface prior to imaging, which could introduce artefacts in membrane topography when cells are landing on top, or 2. Seeding of fiducials after landing of cells, where the majority of fiducials would lay on top of the cells, i.e. outside the investigated volume. Therefore, we established a fiducial free method, which, based on the measured axial localization precision of <20 nm (after drift correction) is, at least as good as commonly reported, fiducial based, methods. We added an according sentence to the manuscript.

3. The dTRABI approach seems a very promising alternative way of acquiring 3D information.

3.1 When the authors irradiate the whole cell in epifluorescence could it lead to preferential bleaching of certain structures over others, has this been accounted for, if so, how?

Authors: We thank the reviewer for raising these questions. The sample is embedded in switching buffer, i.e. an aqueous buffer with a thiol containing reducing agent. Therefore, the overwhelming majority of dyes are not bleached over time but transferred to a meta-stable dark state. The idea of starting in epifluorescence mode is to transfer a huge fraction of dyes within the entire axial range of epi illumination into the non-fluorescent dark state to further minimize out-of-focus contributions, before changing the illumination to HILO. Out-of-focus contribution as stated in comment 2 is very critical in 3D SMLM and could affect the axial localization precision.

We therefore added a sentence in the main manuscript: "Prior to the acquisition, we briefly irradiated the whole cell in epifluorescence to further minimize out-of-focus contributions by transferring the majority of dyes into the non-fluorescent dark state (see Methods)."

3.2 Given that different molecules are expressed at different densities and that the "resolution-enhancement is more pronounced for structures with a medium to high local density" is this easily solved with the quantitative localization analysis, how was this accounted for?

Authors: Every step of our quantitative localization approach takes all detected localizations into account and therefore the full axial distribution was analysed. If there were different expression level - and thus densities - in the axial direction, then this will be reflected by their axial distributions as shown in Fig. 5 with an effect on the metrics such as z-distribution width or peak-

to-peak distance. From a biological point of view, it is important to include this variety into the analysis to distinguish between different mutants; from the technical point of view very high densities can, in principle, lead to imaging artefacts as emitters can overlap and lead to false-positive localizations. Therefore, we carefully adjusted sample preparation, switching buffer composition and illumination intensity during image acquisition to a level that allowed us to image emitter densities suitable for quantitative SMLM. Generally, we chose imaging lengths, i.e. number of acquired frames, to be sufficiently large to oversample the structure (in localization space). This way, the resulting structural mapping is following the underlying protein density and is not affected by local protein density variations.

4. When staining CD4 and CD45 the authors used whole antibodies. Can the authors confirm whether antibody was used on live or fixed cells? This was unclear from the methods section. Furthermore, antibodies can introduce biases in the labelled population due to fixation, epitope, affinity and/or protein density. The authors mention that a CD4 labelled with a fluorescent protein (FP) was used for sorting etc. Can the authors compare the distribution of antibody labelled CD4 with CD4-FP? This is an important point to review and clarify.

Authors: The reviewer is right. Directly-labelled, full-size antibodies were used to visualise CD4 and CD45 at the T-cell surface (except of CD4 in Fig. 7). Labelling was performed on FIXED cells. All antibodies used in our group (Heyrovsky Institute in Prague) are thoroughly tested for their specificity using cells, which do not express the protein of interest and with re-introduced protein of interest. Antibodies probably do not label all molecules at the surface; however, we see a good correlation between GFP signal and signal from antibodies. In all presented experiments, CD4KO-Jurkat cells were used with re-introduced CD4-GFP fusion-protein variants to image CD4 distribution on T cells. Similar results were achieved for endogenous CD4 on native Jurkat T cells. However, CD4 expression on Jurkat cells is unstable and rather low. Labelling of endogenous protein thus did not provide statistically analysable data.

We have revised (expanded) Methods section to clarify labelling strategies and revised figure captions to indicate the labelling strategy there as well.

We have compared CD4-GFP distribution on living and fixed T cells using confocal microscopy (new Supplementary Fig. S7; applies also to Comment 9.). No significant difference in the receptor distribution pattern was observed. We have added a note to the manuscript.

5. In figure 7 where the authors suggest that CD45 might be excluded from CD4 enriched areas all that is presented is 2D SMLM data, do the authors possess now any more information from multi-colour dTRABI analysis? This would be key for the claims in Fig. 7c.

Authors: The reviewer is right that performing 2-colour 3D dTRABI analysis would be ideal to undoubtedly determine relative distribution of CD4 and CD45 in the same cell. Unfortunately, this method is not available to us yet, as any quantitative multi-colour 3D SMLM approach needs delicate calibration and customisation. While our preliminary data were promising at the time of submission of this manuscript, the Covid19 pandemic severely hindered our efforts in finishing this setup.

Nevertheless, we would like to point out that Fig. 7 demonstrates strong segregation of CD4 and CD45 – see also Line profiles, where Green and Magenta signal almost perfectly alternate. Fig. 7d shows an early contact of a cell with glycine-coated surface. Here, almost exclusive accumulation of CD4 in the tips of membrane protrusions is well visible (and highlighted in Line profiles). We have added single channel images to the manuscript to further indicate how different is

distribution of these two receptors in non-stimulated T cells (Fig. 7). Our observation is also in agreement with recently published work of Jung and colleagues (Jung, Wen et al. 2020).

6. It's a bit confusing throughout the paper exactly which reagents were used, i.e., for CD4 there is data with CD4-GFP, CD4-mEOs and direct label with a full antibody against CD4, the manuscript would benefit from a clearer description of the experiments and reagents used.

Authors: We are thankful for this comment. The labelling of diverse molecules was evidently not accurately described throughout the manuscript and this caused confusions with all three referees. We have revised all relevant figure captions and expanded Methods to clarify labelling strategies for each case.

7. In the methods section, Jurkat T cells (clone E6-1 and not E6)

Authors: Corrected.

8. In the Jurkat CD4-KO did the authors test the potential downregulation of Lck, as this could influence the activity of these cells.

Authors: This is an excellent point. Thank you. We have verified Lck expression and found no difference in Lck expression between the source Jurkat T cells and their CD4 KO variants (see below).

Comparison of Lck expression in the original Jurkat T cells (left hand panel) and their CD4 KO variant (right hand panel). The expression was tested using flow cytometry and home-made rabbit anti-Lck polyclonal antibody (kind gift of Vaclav Horejsi, Institute of Molecular Genetics, Prague).

9. Did the authors try any other fixation method? Several reports have claimed that inadequate cell fixation methods could lead to artifactual molecular redistribution by the antibodies themselves. Do the authors have any indication that fixation conditions do not affect the outcome? For example, PFA (paraformaldehyde) vs PFA+GA (glutaraldehyde)?

Authors: We have compared CD4 distribution in living and fixed cells using confocal microscopy (Supplementary Fig. S6). These data demonstrate that fixation does not alter significantly CD4 distribution in non-stimulated T cells. We have added a note to the manuscript. Main consequence of the fixation is reduced number of cells after the first contact with the optical surface (covered with glycine). These are evidently frequently washed off the surface.

We could not use glutaraldehyde in the first fixation step (before labelling; see Methods). GA interferes with antibody labelling. Our protocol involved pre-labelling fixation with PFA only (with 2% sucrose to reduce stress conditions; (Spitaler, Emslie et al. 2006)). There is also a second fixation step after labelling, which includes GA in the fixation solution. We have revised Methods section to clarify our labelling strategies.

Reviewer #2 (Remarks to the Author):

The manuscript by Franke et al. presents an improved SMLM method with axial resolution capable of resolving nanotopography of the basal membrane of adherent cells, and apply it to investigate the organisation of two key proteins involved in activation of T lymphocytes, CD4 and CD45. They improved and simplified their previous SR imaging approach (TRABI), which will help its wider adoption. They also employed an alternative protocol for cell immobilisation, necessary for SMLM, using glycine instead of PLL coating to reduce non-specific activation of T cells and prolong their viability – a welcome step towards controlling experimental conditions in this sensitive application. They finally aimed at elucidating the origin of CD4 clusters in unstimulated T cells.

The manuscript is very well written, clearly structured, and graphical material nicely presented. The methodological contributions of the paper, both regarding imaging modalities and sample preparation, are original and of interest to the wider field. Regarding the specific findings about the organisation of the investigated protein clusters and their biological role, though, I would encourage some further corroboration.

Authors: We are very thankful for positive evaluation of our work and providing several critical comments, which, we believe, led to the improved manuscript.

My main questions and suggestions are the following:

1. In several places the authors claim that “glycine-coated optical surfaces ... preserve the native membrane morphology of immobilized cells”. Without quantification and a proper experimental control (e.g. with cells suspended in a hydrogel, comparison of basal vs apical membrane, or temporal evolution from very early after landing as they attempted qualitatively), though, this should be toned down to comparison with PLL (e.g. “cells flatten less than on PLL”).

Authors: The reviewer is right that, due to technical limitations (see below), we do not know exactly the level, to which glycine on glass coverslips preserves cell surface morphology. We have revised the manuscript in line with the reviewer’s suggestion to tone down the superiority of glycine coating and specifically highlight only on its advantages when compared to PLL. We apologise but the comparison of our samples with cells in hydrogel is impossible. As mentioned in the Discussion, cell positioning far above the optical surface (hydrogel) is not compatible with dTRABI method. Similarly, the data from the contact site cannot be compared with those from the upper surface of the cell using 3D SMLM. However, our new confocal and TIRF microscopy data indicate that large protrusions (detectable by these methods) are still detectable in cells on glycine 10 min after loading of cells to the imaging chamber. This is much longer compared to the cells on PLL coated coverslips (new Fig. 1c,d,e; Supplementary Fig. S1). These data, together with new IRM analysis (Supplementary Figs. S4), strongly support our statement that glycine-coating of coverslips prevents (for 2-10 min) large collapse of surface structures, including membrane protrusions.

2. If there is any other way of characterising the mechanical properties of the glycine layer, that would be very helpful to the community.

Authors: We agree with the reviewer that further characterisation of glycine layer would be helpful. Unfortunately, we are not aware of any technique, which can provide more information about this material. We would be very thankful if the reviewer can point to such a method. AFM did not perform well with this super-soft material.

3. The authors thoroughly analyse the distributions of axial localisations, approximating them with bi-Gaussian functions and comparing the extracted parameters. However, I find some of the descriptors questionably informative, especially when the second component is small or largely overlapping with the first (Fig S6a). Would it perhaps help to fix one component (main peak) and then fit only the shoulder, which varies most between the samples? Or perhaps complement z_w with some interquantile distance?

Authors: We in principle agree with the reviewer, that the shoulder is the most important component of the majority of distributions. The suggested method to fix the main peak and only fit the shoulder would be possible, but we also see data with clearly two separated Gaussian distribution, where no shoulder could be characterized (see Supplementary Fig S10. Segmentation of cells). Therefore, we decided to use a bi-Gaussian function and metrics such as z-distribution width and peak-to-peak distance.

We also agree with the reviewer, that one could extract other or additional metrics from the fits, but we found, that the three main metrics reported in the manuscript are sufficient to characterise the data distinctly. Possibly, the addition of other metrics can be beneficial for the analysis of different structural data in the future.

4. The reader remains hungry for further insights into the meaning of the devised proteins' axial distributions, e.g. on the origin of the two populations (localisations on separate structures/folds of the membrane?). Please discuss also to what degree this will be relevant to a lymphocyte interacting with another cell with a less flat surface.

Authors: It seems highly probable from the literature that lymphocytes interact with other cells via membrane protrusions, specifically via microvilli (Fisher, Bulur et al. 2008, Cai, Marchuk et al. 2017, Kim, Mun et al. 2018). However, our data do not provide enough information about physiological interactions of T cells. Therefore, in agreement with the other two referees of the manuscript, we would like to avoid further biological conclusions based on the limited data, which were used to highlight the advantages of our novel methodological approach.

5. Central regions of the cells in dTRABI images of CD45 and CD4 look very "flat". Please discuss the potential effects of the localisation filtering on biasing the results. It would be instructive to compare raw total intensities to those generated from the reconstructed images after filtering, to see if any "features" are systematically filtered out (by intensity, z , etc).

Authors: As mentioned above, cells are spreading also on glycine-coated coverslips after longer incubation. This may lead to partial collapse of membrane morphology in the central part of the cell (effect of nucleus??) even at the time points, which were used for landing in our experiments (followed by rapid fixation at 37 gr C). Unfortunately, we cannot synchronise landing of cells on the surface. Thus, there is high diversity between the cells with respect to the time after the first contact with the coverslip. This is probably the cause of flattened central area in some cells.

Nevertheless, despite these inherent heterogeneities our quantitative analysis shows a distinct difference between sample types.

dTRABI has an effective axial working range of 1 μm , due to the used biplane configuration (Franke, Sauer et al. 2017). Therefore, we believe that we already show unfiltered images in the common SMLM sense, since the observed membrane structures are expected to be less in size. The only relevant applied filters are a low intensity threshold and a high upper limit for the allowed FWHM of spots within the localisation analysis (both stated in Methods, *low* and *high* compared to commonly reported values). Both parameters, in conjunction with the HILO illumination and the focal plane, were intentionally chosen to reproduce the entire structure of the touching plasma membrane.

6. From the presented dTRABI images of CD4 CS1 cells (Fig S6 & S9), it seems like 3 out of 5 are very similar to the WT. A comparison of cell-by-cell distributions (instead of ROI-wise) would therefore also be informative. Can this mutation influence the attachment/spreading of the cell?

Authors: We agree that some images of the receptor distribution for CD4-WT and CD4-CS1 variants may appear similar. This is primarily caused by the different expression levels between individual cells caused by transient transfection. A broad spectrum of expression levels was tested for each condition (receptor). Only cells with extremely high (> 1.500.000 localisations) or low (<250.000 localisations) expression were excluded from the analysis. Closer inspection of CD4-CS1 cells with lower expression indicates that this variant is also present in the plasma membrane base, not exclusively on the protrusions (as CD4 WT) – see an example below.

Magnified example of Jurkat T cell expressing lower levels of CD4-CS.

Individual cells were not compared quantitatively due to varying levels of receptor expression. We have used cells with different expression levels in our analysis (see above). The time from the first contact of a cell with the surface also affects its morphology and this effect varies between cells (see response to Comment 5). Only full quantitative data from all cells were used for a direct comparison of different receptors or their variants.

The visual 'similarity' of the receptor distributions further highlights the importance of quantitative analysis, which we have performed on all cells. Our biological conclusions are based on quantitative image analysis and not visual inspection of images.

7. Can the observed exclusion of CD45 from the CD4-rich zones be explained also by steric effects at the contact zone with the surface, not (solely) from different distributions along the protrusions? To support authors' interpretation, in my view, the case in Fig 7d-e (cell just landing) is much more compelling than the case in Fig 7a-b (fully spread cell) where distribution of CD4 seems very different from other images with dTRABI.

Authors: We do not think that exclusion of CD45 from CD4-rich zones is caused by the steric effect (only). CD4-rich zones are well penetrated by anti-CD4 antibodies, which have similar size when

compared to CD45 extracellular domain. Also, the size of CD4 does not differ that much from CD45 as, for example, in the case of CD2 or TCR (in membrane normal orientation).

The reviewer is right that images of CD4/CD45 distribution differ between each other. We expect that, with time, membrane protrusion collapse under the cell body (Supplementary Fig. S15), which may cause a larger overlap in the distributions of CD4 and CD45. However, the distribution of CD4 in Fig. 7a,b does not differ significantly from the 2D projection of 3D TRABI data presented in Supplementary Fig. S11.

8. In the Discussion, please compare dTRABI to other SMLM (and other SR) approaches with sub-diffraction resolution (nicely listed in the Introduction), to help readers decide which one suits their application best. Please also discuss potential limitations and biases of the SMLM approach, requiring long acquisition and immobilisation, e.g. detecting stationary vs mobile protein clusters, potency to induce clusters by immobilisation, etc.

Authors: We thank the reviewer for encouraging us to review SR techniques and their peculiarities in context of our work. However, there already exists an extensive literature on the merits and limitations of different SR approaches for a variety of cell biological questions (e.g., (Schermelleh, Ferrand et al. 2019)) and we politely decided to not revisit this particular discussion, since we feel that this would be outside the scope of our manuscript. We do not claim universal superiority of dTRABI over other SR methods.

As for the suggestion to discuss the limitations and biases of SMLM in context of e.g., clustering artefacts, we also do feel that there has been quiet a vivid and published discussion within the SMLM field already, to which we refer to in the introduction of the manuscript. We feel, that in the context of our work, there are no additional biases or potential problems that have not been discussed elsewhere already. We made great efforts to take the current knowledge on how to avoid any form of imaging, labelling and fixation artefacts into account.

Finally, we would like to point out that dTRABI is an extension on the original TRABI method and therefore has to be seen within the group of multi-plane SMLM approaches. We conclusively showed in the original TRABI manuscript that our photometric approach is superior to classical biplane (here we show, that dTRABI is even better than TRABI), which, in turn, has been characterised extensively as the best 3D SMLM method in terms of theoretical, minimum localisation precision, compared to the methods such as astigmatism.

Based on the results presented in the manuscript, we believe that our workflow, combining glycine coating of coverslips and dTRABI, is the best approach to date to study 3D nanoscale organization of the plasma membrane. Moreover, we believe that the same could be true for many other biological questions, but we do not doubt that for many other problems, different SMLM or SR approaches might be better suited.

9. Abstract: “near isotropic 3D nanometer resolution” – please specify (e.g. “resolution around A/B in XY/Z on such and such samples”)

Authors: Lateral and axial localization precision were <10 and <16 nm (standard deviations, see Supplementary Fig S8. Localization precision), respectively. Therefore, we achieve a near isotropic resolution of better than 38 nm (FWHM + 16 nm × 2.355).

10. Fig 1e: please describe in the caption whether each square represents a single cell or a repeated experiment (with N cells)

Authors: The manuscript was revised accordingly.

11. Fig 3: the corrected images in panel a and b are the same, which is illustrative, but confusingly synthetic (apparent need for just one OR the other correction on the same dataset). Please clarify, e.g. indicate the order of corrections in a real workflow.

Authors: We adjusted the respective methods paragraph and figure caption according to the reviewer's suggestion to clarify the procedure.

12. Fig 5: where is the $z = 0$ plane in the distributions?

Authors: In many cases, the plane with $z=0$ can be below the optical surface. The position of this plane depends on the focus.

13. Fig 5: are panels h-i missing or is the caption redundant?

Authors: Panels h and i are part of Fig. 5 (right-hand side, top part). The position is not perfectly logical due to the flow of the main text. In fact, if the reviewer has a good suggestion on how to improve on the current layout, we would adapt the figure accordingly.

14. Fig 6: what exactly does "lacking CD45" refer to?

Authors: We would like to thank the reviewer for this comment. Yes, the title did not sound expertly. The text was revised to better describe the phenomenon.

15. Fig S5: Do you report lateral or axial resolution on these 3D data sets?

Authors: The reported values describe the 2D FRC resolution. We adapted the figure caption to clarify.

16. Page 22, T cell viability assay, line 13: "fist" -> "first"

Authors: Corrected.

Reviewer #3 (Remarks to the Author):

In their current ms. Franke et al. focus on the localization of membrane proteins in three dimensions below the diffraction limit of visible light. More specifically, they showcase two approaches to reach that goal: The first deals with glycine coating of glass surfaces as a means to provide a substrate for cell attachment without inducing substantial changes in cellular morphology (i.e. loss of microvilli and other membrane protrusions). The second concerns improving existing single molecule localization microscopy (SMLM) methodologies for the third dimension with the use of highly inclined and laminated optical sheet illumination (HILO). The motivation here is to reduce cellular background by exciting predominantly cellular membranes in contact with the glass-surface preparation (a requirement for single molecule fluorescence detection) yet not limiting fluorescence to 100-200 nm into the specimen, as would be the case sample excited via evanescent light resulting from Total Internal Reflection modalities. To make their case, the authors image CD45 and CD4 expressed on Jurkat T-cells. They provide data suggesting differential membrane localization with regard to membrane protrusions: While CD4 but not its palmitoylation-deficient appear closer to the glass surface, CD45 seems at least in part be further away from the glass plane. Given the undeniably important implications of membrane topology and adequate quantitation for understanding cellular

function I am highly in favor of exploring new ways and enhancing already existing methods to conduct 3D superresolution microscopy. Yet at the current stage I consider the study not sufficiently compelling to warrant publication in Communications Biology.

Authors: We thank the reviewer for evaluation of our work and for providing several critical comments, which we used to improve the manuscript. We hope that, in its current state and after responding to the reviewer's comments, the manuscript is more attractive for the readers of Communications Biology.

Reasons for my overall assessment are stated below:

1. Overall presentation. I find both the introduction and the discussion section redundant and not written in a concise fashion. A lot of text deals with issues that have no direct relevance for the data shown. What exactly is the narrative? Is it biology or methodology? If it is the first, the message is too limited, yet the methodological part receives too much attention. If it is the latter, then why spend much effort dealing with potential (yet not proven) biological implications? Instead, showing more evidence for method robustness and integrity would help. Figures 2, 3 and 4 could be combined to one figure. The same is true for figures 5 to 7.

Authors: The message of this work is more methodological than biological. However, we have made several biological observations, which are discussed throughout the manuscript. These data are partially novel and partially in line with the literature published very recently (Razvag, Neve-Oz et al. 2019, Ghosh, Di Bartolo et al. 2020, Jung, Wen et al. 2020). We also believe that direct biological application of our novel methodology strengthens its attractiveness for potential users.

We have revised biological parts in the Introduction and Discussion, as suggested by the reviewer. Also, we have added several pieces of evidence to further support the strength of our methodology (see below).

There are statements I find close to overstating.

Authors: We apologise if our statements seem not to be supported by the results. However, to avoid such cases, we need to be pointed to the specific sentences or sections.

I am moreover of the opinion that in modern (and space restriction-free) days of online publishing methodologies should allow for easy data reproduction without the need for consulting referenced papers. This is clearly not the case here.

Authors: The methodology section, which we believe was already detailed, was further expanded to avoid any lack of information (see our responses to more specific points of all three referees).

Furthermore, it took me a while to find out the specifics of the staining / visualization procedures employed for CD4 (antibody or genetic GFP fusions). More clarity in the results and methods section would help.

Authors: We agree with the reviewer that the staining strategies have not been accurately described in figure captions of the original manuscript. The relevant parts have been revised to state unambiguously the labelling. The methods have been revised to provide more information and clarity.

2. Glycine-coating. This is an interesting aspect of the ms., which would in my view clearly benefit

from providing answers to a number of questions: How mobile are Jurkat T-cells on such surfaces? Do they crawl and scan their environment or do they stay put?

Authors: This is an interesting point. We have characterised mobility of Jurkat T cells on glycine-, PLL- and serum-coated coverslips. The data were added to the Supplementary Information of the manuscript (Supplementary Movies 3-5). The main text of the manuscript was modified to report on these, in our eyes, interesting data. In brief, as expected, Jurkat T cells are slowly rolling on serum-coated coverslips. Such surface does not allow for fixation of cells and SMLM imaging. On glycine surface, cells stop rolling early after the first contact but continue to screen local environment by generating (and collapsing) numerous protrusions for the whole period of imaging (15 mins). Similarly, cells stay put immediately after the contact with the PLL-coated surface but their ability to dynamically screen the environment is reduced compared to glycine-immobilised cells. These two surfaces are applicable for SMLM imaging.

2.1 Do primary T-cells attach to such surfaces?

Authors: Yes, all tested cells (including B and T cells, adherent cells, neuroblastoma cell lines) attach to glycine-coated surfaces. We have added images of cell contacts with the glycine-coated surface for human B cell line (Raji), murine macrophage-like RAW 264.7 cells and Rhesus monkey kidney fibroblast (COS-7) to the Supplementary Information of the manuscript – Supplementary Fig. S5. Images of primary cells could not be presented in this work. We do not have access to primary cells - it requires specific agreement from our Ethical Committee for each project. We were informed by our collaborators from the Main Hospital in Prague – Motol (Prof. Ondrej Hrusak) that primary T and B cells also attach to this surface.

Examples of human Raji B cell line, murine RAW264.7 macrophages and Rhesus monkey COS-7 fibroblasts immobilised on glycine- and PLL-coated coverslips. Left panel shows the contact site of a cell with the optical surface. Right panel shows the cell approximately 1 μm apart from the optical surface.

We also provide images of cell contacts with PLL-coated coverslips. These data further support the fact that glycine-coated coverslips better preserve surface morphology of diverse cells compared to PLL coating.

2.2 I understand the value of imaging T cells in a resting state (prior to antigen recognition), but what would this form of attachment represent in the real world (resting T cells reside in different microenvironments such as skin, other organs, secondary lymphatics, blood, endothelial cells etc...). I feel categorizing glycine attachment as the most physiological for the preservation of filopodia is subjective and not necessarily true. Regardless, such filopodia are inherently interesting and deserve investigation, and here glycine coating may make the difference. But clarification in the text would position the use of glycine in a more adequate context.

Authors: The reviewer is right that glycine gel-like surface does not represent physiological environment for T cells (or any other cells). In this work, we present a novel surface for high-resolution 3D SMLM imaging, which requires immobilisation of cells, positioning close to the optical surface and low background. These pre-requisites were discussed in the original manuscript. To date, suspension cells used for 3D SMLM imaging were immobilised on PLL-, antibody- or ligand-coated flat surface. We provide an alternative, which better preserves surface morphology and viability of cells compared to the currently available methods. We have expanded the manuscript to support this statement with more data (Fig. 1 and Supplementary Figs. S1, S2-S6). The text was also revised to better describe the position of glycine with respect to cell physiology.

In non-motile cells, membrane protrusions are not in the form of filopodia, but rather of ruffles and microvilli (Majstoravich, Zhang et al. 2004, Fisher, Bulur et al. 2008, Cai, Marchuk et al. 2017, Kim, Mun et al. 2018).

2.3 How about other cell types? Dendritic cells, B cells, macrophages? Do they attach? A methodology paper should address this and a simple experiment would tell immediately.

Authors: We agree with the reviewer. Please, see the response to the comments 2.1 and 2.2 above. B cells and macrophages behave in a similar way to T cells. Cells with more complex morphology (e.g., dendritic cells) will require longer time to attach to this surface.

3. Choice of T-cells: Jurkat T-cells. I surely understand the practical benefits of using Jurkat T-cells, and obviously, to make a methodological point, the use of Jurkats may suffice. However, Jurkats are far removed from primary T-cells with regard to their physiology, and it shows: Primary T-cells do not undergo apoptosis when in contact with poly-L-lysine coated surfaces, nor do they flux calcium. The same is true for the use of protein-functionalized planar glass-supported lipid bilayers. Primary T-cells require antigen for activation without exception (while Jurkats flux spuriously). The operational need for Jurkats has vanished progressively with lenti-/retroviral expression systems and the emergence of CRISPR-CAS9. I hence find their use irrelevant if not at times misleading for drawing biological conclusions.

Authors: As stated above, this is mainly methodological paper. Therefore, we have used Jurkat T cell line to provide extended information about the novel method. The reviewer is right that Jurkat T cells are far from being similar to primary T cells. But primary cells are not accessible for many laboratories, especially those with advanced imaging and biophysics expertise. The use of primary cells requires approval of ethical committee and patient's informed consent. We have limited access to primary immune cells.

The use of lenti-/adenoviral vectors is also not universally available for imaging laboratories (e.g., requirement for a higher biohazard level). Therefore, genetic modification of cells in many laboratories is limited to standard transfection methods, which are poorly efficient in primary T cells.

We agree that Jurkat T cells flux calcium in a less controlled way compared to primary T cells. However, primary T cells flux calcium on PLL (Santos, Ponjavic et al. 2018). We do not know yet, if the viability of primary immune cells is better on glycine surface compared to PLL. This needs to be experimentally verified in the future. But we believe that since this is the first work using glycine coating of glass coverslips for imaging, it provides enough information to stimulate further research, including that on primary cells.

4. CD4 mutant. What is the evidence that the palmitoylation-deficient mutant is effectively transported to and predominantly present on the cell surface? For immunofluorescence on fixed but impermeable membranes this is not an issue, yet for any approach involving XFP(mEos/psCFP2 etc.)-gene fusions this may become an obstacle, in particular in view the HILO illumination scheme. An alternative explanation for locating this mutant further away from the glass surface may indeed be its association with internal membrane compartments. For the palmitoylation-deficient mutant of the Linker of Activation of T cells (LAT) this is certainly the case and has led to wrong conclusions in the past. The paper cited by the authors to support their claim that CD4 is clustered (while the mutant is not) is based on the use of CD4-fusion proteins.

Authors: Palmitoylation-deficient mutant is efficiently delivered to the surface as reported in our previous work (Lukes, Glatzova et al. 2017). Indeed, confocal images of the native CD4 and its non-palmitoylatable mutant are almost indistinguishable (see images below).

Confocal microscopy of CD4 WT-GFP transiently expressed in Jurkat cells (Glatzova, 2013; Master thesis, Charles' University, Prague).

Confocal microscopy of CD4 CS-GFP transiently expressed in Jurkat cells (Glatzova, 2013; Master thesis, Charles' University, Prague).

We have revised the text of the manuscript to clarify this issue.

In Figures 5, S10, S11 and S13, we have used antibody labelling in the absence of cell permeabilization to study exclusively the surface distribution of CD4 CS mutants. All these data indicate that surface expression of CD4 is not controlled by its palmitoylation.

We agree that palmitoylation mutant of LAT is not expressed at the cell surface. Actually, we studied this issue in a very comprehensive work (Chum, Glatzova et al. 2016).

4.1 The paper cited by the authors to support their claim that CD4 is clustered (while the mutant is not) is based on the use of CD4-fusion proteins.

Authors: The reviewer is right that CD4 clusters in our previous work were studied using GFP/mEos2 fusion proteins. However, our preliminary 2D SOFI data indicate that endogenous CD4 is present exclusively in 'clusters' in primary T cells. The data of other groups (for example: (Ghosh, Di Bartolo et al. 2020)) support this view.

5. SMLM. Excessive blinking (for PALM and dSTORM) as well as partial transfer into a reversible dark state (dSTORM) renders SMLM vulnerable to cluster artefacts.

Authors: We agree with the reviewer, in that SMLM can result in artefacts, such as artificial clusters, if not performed carefully or with non-ideal dyes.

In part depending on their chemical nature a few individual fluorophores give rise to unpredictable blinking which cannot be accounted for by simple merging algorithms. To make things more complicated, only a small subset of originally available fluorophores may indeed be available for dSTORM (following irreversible ablation of the majority of fluorophores) producing an exaggerated pointillistic image with additional cluster artefacts. This needs to be addressed.

Authors: We agree with the reviewer, in that, photo-blinking of the sample is a spatio-temporal stochastic process, the differentiation and grouping of individual localisations is an ambitious task and structural under-sampling due to photo-bleaching is a common problem in fluorescence microscopy. However, since we do not aim to provide cluster specific localisation counts, grouping of localisations are not part of the manuscript. We are using Alexa Fluor 647 as our main reporter, a carbocyanine dye which is known to act as reliable photoswitch with several tens of photo-cycles under our switching buffer conditions (Heilemann, Margeat et al. 2005, Dempsey, Vaughan et al. 2011, van de Linde, Loschberger et al. 2011). The same is true for mEos as our secondary reporter,

thus photo-bleaching is not a limiting factor for structural resolution in our experiments (Endesfelder, Malkusch et al. 2011). Furthermore, we took great care to adjust buffer, imaging, and analysis parameters to avoid any density-based localisation artefacts in our data. Please, also refer to our answer to Reviewer 1 Comment 3.2 for further details.

A simple way could be to take an image of the cell surface prior to bleaching/photoswitching and compare this image with a SMLM image which has been computationally convolved / blurred using the FWHM employed for localization (i.e. reversing the SMLM process). If the SMLM image is correct, the intensity distribution of the convolved image matches that of the first ensemble image taken.

Comparison of fluorescence and SMLM image is indeed an important parameter to evaluate the reliability of novel photoswitches. For the dye used in this work, Alexa Fluor 647, this has already been done to great extent (cf. (Bates, Huang et al. 2007, Heilemann, van de Linde et al. 2008, Dempsey, Vaughan et al. 2011, van de Linde, Loschberger et al. 2011)). Due to the carefully studied photoswitching kinetics it can be summarized that Alexa Fluor 647 and Cy5 belong to the most established and best performing dyes in dSTORM imaging. Although we are aware of alternative approaches for the assessment of super-resolution image quality based on diffraction limited images, e.g. (Culley, Albrecht et al. 2018), we do not agree, that in case of our data, this kind of analysis would be able to indicate the validity or quality of the SMLM data. Blurring of pointillistic nanoscale clusters of high density would actually lead to a rather homogeneous local 'fluorescence' signal, the same would be true for homogeneously distributed localisations. Additionally, we do not report a mere pointillistic clustering of surface receptors, but rather a structured accumulation, or lack thereof, of proteins at the nanoscale. In addition, the general geometry and structure of CD4 accumulation was consistent in several different imaging modes throughout the manuscript (confocal, SOFI, 2D SMLM, 3D SMLM).

6. Main biological claim of the ms. that CD4 and CD45 segregate to different sites on microvilli. The evidence for lack of colocalization (shown in figure 7; a larger representation would help) fails to convince me. If this were the case, microvilli displayed in X-Y would clearly show this behavior, but they do not. The tips are never green (i.e. CD4 positive but CD45 negative) but always white (positive for both). Also, if conclusive the single color analysis provided in figure 4 would rather contradict such a scenario. However, the results shown in figure S9 let me question the validity of comparisons which are based on single color detections, as there appears to be considerable variability among different cell contacts. This leads me to conclude that dual-color colocalization is the only reliable means to derive topographical differences.

Authors: The reviewer is right that there is a large variability in contact geometry between individual cells. Therefore, single colour 3D superresolution images are not sufficient for comparison of the receptor distributions. It is the quantitative approach introduced in this work, which helps to distinguish diverse distributions of surface receptors before the availability of a reliable and user-friendly multichannel 3D methods with sufficient resolution. Unfortunately, microvilli (in contrast to filopodia) are not large enough to be visualised with resolution-limited methods.

Newly added single colour data (Fig. 7), together with line profiles (see middle panel of Fig. 7), demonstrate that CD4 accumulates at the tips of detected protrusions (microvilli and ruffles), while CD45 is excluded from these parts. It is possible to distinguish the position of the two receptors only in the protrusions, which extend to the free space surrounding the cellular

footprint in this 2D projection (Fig.7d). Axial overlap of CD4 at the tip and CD45 at the shaft of microvilli is probably responsible for white colour in the central part of the landing cell in Fig. 7d.

Our conclusions are further supported by recently published work (Jung, Wen et al. 2020).

7. Validity of the 3D-nanotopography. Evidence that is independent of the showcased method would be helpful and compelling and should be in reach. Because the large majority of the 3D-structures observed in the SMLM images are large enough to be resolvable above the diffraction limit, a convincing case could be made by comparing the 3D superresolution plots with diffraction-limited RCM images taken from the same contacts. Both methods are sensitive within similar Z-ranges.

Authors: The reviewer is right that interference reflection microscopy (IRM) can provide some evidence about the 'roughness' of a cell contact with optical surface. We have thus imaged living Jurkat T cells on glycine- and PLL-coated coverslips by IRM. Acquired IRM images demonstrate that cellular contacts with glycine-coated surface are more geometrically complex than for the cells on PLL. This applies to both evaluated times - 5 and 10 minutes after landing (from the addition of cells to the imaging chamber). We have added four representative examples of IRM images for all cases to the Supplementary Fig. S4 and discuss these data in the main text of the manuscript (see figure below). We believe that these data (together with those in the original manuscript) sufficiently support the evidence that glycine-coating of coverslips better prevents collapse of cell surface morphology compared to PLL.

Glycine

PLL

5 min

10 min

Interference reflection microscopy (IRM) of contacts formed by cells landing on glycine- (left side) or PLL-coated (right side) coverslips. Four examples are shown for each time point (5 and 10 min) and surface coating.

References:

- Bates, M., B. Huang, G. T. Dempsey and X. Zhuang (2007). "Multicolor super-resolution imaging with photo-switchable fluorescent probes." Science **317**(5845): 1749-1753.
- Cai, E., K. Marchuk, P. Beemiller, C. Beppler, M. G. Rubashkin, V. M. Weaver, A. Gerard, T. L. Liu, B. C. Chen, E. Betzig, F. Bartumeus and M. F. Krummel (2017). "Visualizing dynamic microvillar search and stabilization during ligand detection by T cells." Science **356**(6338): 598.
- Cebecauer, M. and R. Šachl (2016). Lipophilic Fluorescent Probes: Guides to the Complexity of Lipid Membranes. Fluorescent Analogues of Biomolecular Building Blocks. Design and Applications. M. Wilhelmsson and Y. Tor. Hoboken, New Jersey, John Wiley & Sons.: 367-392.
- Chum, T., D. Glatzova, Z. Kviclova, J. Malinsky, T. Brdicka and M. Cebecauer (2016). "The role of palmitoylation and transmembrane domain in sorting of transmembrane adaptor proteins." J Cell Sci **129**(1): 95-107.
- Culley, S., D. Albrecht, C. Jacobs, P. M. Pereira, C. Leterrier, J. Mercer and R. Henriques (2018). "Quantitative mapping and minimization of super-resolution optical imaging artifacts." Nat Methods **15**(4): 263-266.
- Dempsey, G. T., J. C. Vaughan, K. H. Chen, M. Bates and X. Zhuang (2011). "Evaluation of fluorophores for optimal performance in localization-based super-resolution imaging." Nat Methods **8**(12): 1027-1036.
- Endesfelder, U., S. Malkusch, B. Flottmann, J. Mondry, P. Liguzinski, P. J. Verveer and M. Heilemann (2011). "Chemically induced photoswitching of fluorescent probes--a general concept for super-resolution microscopy." Molecules **16**(4): 3106-3118.
- Fisher, P. J., P. A. Bulur, S. Vuk-Pavlovic, F. G. Prendergast and A. B. Dietz (2008). "Dendritic cell microvilli: a novel membrane structure associated with the multifocal synapse and T-cell clustering." Blood **112**(13): 5037-5045.
- Franke, C., M. Sauer and S. van de Linde (2017). "Photometry unlocks 3D information from 2D localization microscopy data." Nat Methods **14**(1): 41-44.
- Ghosh, S., V. Di Bartolo, L. Tubul, E. Shimoni, E. Kartvelishvily, T. Dadosh, S. W. Feigelson, R. Alon, A. Alcover and G. Haran (2020). "ERM-Dependent Assembly of T Cell Receptor Signaling and Co-stimulatory Molecules on Microvilli prior to Activation." Cell Rep **30**(10): 3434-3447 e3436.
- Heilemann, M., E. Margeat, R. Kasper, M. Sauer and P. Tinnefeld (2005). "Carbocyanine dyes as efficient reversible single-molecule optical switch." J Am Chem Soc **127**(11): 3801-3806.
- Heilemann, M., S. van de Linde, M. Schuttpelz, R. Kasper, B. Seefeldt, A. Mukherjee, P. Tinnefeld and M. Sauer (2008). "Subdiffraction-resolution fluorescence imaging with conventional fluorescent probes." Angew Chem Int Ed Engl **47**(33): 6172-6176.
- Jung, Y., L. Wen, A. Altman and K. Ley (2020). "CD45 pre-exclusion from the tips of microvilli establishes a phosphatase-free zone for early TCR triggering." bioRxiv (Preprint) DOI: **10.1101/2020.05.21.109074**.
- Kim, H. R., Y. Mun, K. S. Lee, Y. J. Park, J. S. Park, J. H. Park, B. N. Jeon, C. H. Kim, Y. Jun, Y. M. Hyun, M. Kim, S. M. Lee, C. S. Park, S. H. Im and C. D. Jun (2018). "T cell microvilli constitute immunological synaptosomes that carry messages to antigen-presenting cells." Nat Commun **9**(1): 3630.
- Lukes, T., D. Glatzova, Z. Kviclova, F. Levet, A. Benda, S. Letschert, M. Sauer, T. Brdicka, T. Lasser and M. Cebecauer (2017). "Quantifying protein densities on cell membranes using super-resolution optical fluctuation imaging." Nat Commun **8**(1): 1731.
- Majstoravich, S., J. Zhang, S. Nicholson-Dykstra, S. Linder, W. Friedrich, K. A. Siminovitsh and H. N. Higgs (2004). "Lymphocyte microvilli are dynamic, actin-dependent structures that do not require Wiskott-Aldrich syndrome protein (WASp) for their morphology." Blood **104**(5): 1396-1403.
- Razvag, Y., Y. Neve-Oz, J. Sajman, O. Yakovian, M. Rechtes and E. Sherman (2019). "T Cell Activation through Isolated Tight Contacts." Cell Rep **29**(11): 3506-3521 e3506.
- Santos, A. M., A. Ponjavic, M. Fritzsche, R. A. Fernandes, J. B. de la Serna, M. J. Wilcock, F. Schneider, I. Urbancic, J. McColl, C. Anzilotti, K. A. Ganzinger, M. Assmann, D. Depoil, R. J. Cornall, M. L. Dustin,

D. Klenerman, S. J. Davis, C. Eggeling and S. F. Lee (2018). "Capturing resting T cells: the perils of PLL." Nat Immunol **19**(3): 203-205.

Schermelleh, L., A. Ferrand, T. Huser, C. Eggeling, M. Sauer, O. Biehlmaier and G. P. C. Drummen (2019). "Super-resolution microscopy demystified." Nat Cell Biol **21**(1): 72-84.

Spitaler, M., E. Emslie, C. D. Wood and D. Cantrell (2006). "Diacylglycerol and protein kinase D localization during T lymphocyte activation." Immunity **24**(5): 535-546.

van de Linde, S., A. Loschberger, T. Klein, M. Heidebreder, S. Wolter, M. Heilemann and M. Sauer (2011). "Direct stochastic optical reconstruction microscopy with standard fluorescent probes." Nat Protoc **6**(7): 991-1009.

Welliver, T. P., S. L. Chang, J. J. Linderman and J. A. Swanson (2011). "Ruffles limit diffusion in the plasma membrane during macropinosome formation." J Cell Sci **124**(Pt 23): 4106-4114.

Reviewers' comments:

Reviewer #1 (Remarks to the Author):

Thank you to the authors for answering most of the questions and for the manuscript changes that have increased its quality. We can divide the manuscript's content into three main topics, the use of glycine surface as a non-activatory surface, the development of the dTRABI method and the biological observation about CD4 and CD45 distribution at T cell protrusions on a resting state. I still have, however, a few significant concerns/comments regarding two of the topics that are as listed.

- Even if I understand the justification for my previous 1.3 and 1.4, I still believe that in order to confirm that the glycine surface is inert, this needs to be extensively characterised. There are many options to measure calcium release, including imaging-based approaches. Several groups have reported calcium release using confocal measurements, which can easily measure calcium dye fluorescence for hundreds of cells and record timelapse. This "activating" vs "inert" surface question is not critical for the establishment of dTRABI, however, all the biological conclusions made at the end of this manuscript are based on the assumption that glycine surfaces are inert. For example, in the new IRM data, the authors could label cells with calcium dye and follow calcium release as cells interact with both surfaces. This experiment would strengthen the IRM data and confirm that glycine is not simply slowing down contacts, for example (imaging data variability?). To make a clear case for the inert nature of the glycine surface, the authors should try to quantify calcium release over time, especially in the first 10-20 minutes (to fit with the immobilisation and fixation of cells described for SR imaging).

- In Fig. 5, authors claim "strong tendency of wild-type CD4 to accumulate at one specific topographical level". Based on 5f, CD45 colocalises nicely with CD4 in these protrusions. The authors state "broader axial distribution of CD45 receptors than CD4 WT" but not lack/exclusion of CD45 at the tips. This observation seems to be in contrast with the claims/conclusions from the data shown in Fig.7. In Fig. 7, both proteins should be imaged with FP or antibody (as in Fig. 5). Keep in mind that CD4 is shorter than CD45. On top of that, CD4 is labelled with FP and CD45 with antibody, which could exacerbate minor differences in distribution. Differences in labelling approaches could justify why the data shown in Fig. 5 and Fig. 7 is somewhat different, and this variability should be addressed more carefully.

- It is essential to discuss the possibility of CD45 being sterically excluded from glass/glycine-cell interfaces. One way to exclude the role of the extracellular region would be to take advantage of CD45 mutants with truncated extracellular domains, which seem to penetrate surface-cell contacts (described in Chang et al.).

All in all, while the dTRABI approach seems very interesting and has the potential to become a useful method in the field, the data that supports the conclusions for the two biological claims – activating vs inert surfaces and CD45/CD4 distribution – could greatly benefit from relatively simple but important additional experiments.

Reviewer #2 (Remarks to the Author):

I thank the Authors for carefully revising the manuscript. In the letter they have answered most of the points I raised, but I miss their incorporation in the revised work, specifically the following:

Ad 2: One could try Quartz crystal microbalance with dissipation monitoring.

Ad 3: While the cases with two overlapping components (e.g. in Fig S10c ROI#20) reflect in a low p-p and z_w values, the presentations should also highlight the significance of the second component (e.g. the relative fraction f_2 is tiny in Fig S10c ROI#6), which is a key piece of information for the interpretation of the underlying distributions. Probably the easiest and most illustrative way would be to plot localisation counts (as Fig 5b,d,i) averaged over all ROI (aligned

and normalised, as Authors deem appropriate). Alternatively, average f_2 per bin could be plotted as the second axis alongside the parameter histograms, or used to weigh the individual data contributions to these histograms. Please also discuss this in the text to complement the informativeness of all parameters. The text should help readers make sense of the data and numerical results, as nicely exemplified in some places ["... signifying a strong tendency of wild-type CD4 to accumulate at one specific topographical level"], but missing in others [e.g. "Furthermore, the axial analysis of localizations revealed a more pronounced mean value of D_{FWHM} for CD4 CS1 (88 nm) than for CD4 WT (57 nm)"].

Ad 5: After ["As a consequence, many localizations given by the localization software were rejected by TRABI."] and similar statements, referring to the original method, the reader does not easily take home the message on dTRABI improvement on this issue. To me this is important and deserves better exposure in the text.

Ad 6: I fully appreciate the efforts of the Authors for rigorous quantitative image analysis, which should also result in more informed visual inspection. For meaningful interpretation and therefore better insights into biology, I think it would be beneficial to understand whether the differences between the samples come primarily from the overall cell-by-cell variations or from a small subset of cells. From the Authors' response I cannot see why ROIs can be quantitatively compared, but cells (sets of ROIs) cannot, so please discuss this in the text. Please also discuss the effects of different expression levels in the text.

Ad 7: Please incorporate these important points in the discussion.

Ad 9: These numbers are definitely more informative than "nanometre resolution", stated in the abstract – please better specify it there.

Reviewer #3 (Remarks to the Author):

In its current version the ms. addresses almost all issues I have raised previously. There are however 2 aspects that do not.

1. Jurkat T cells are not T cells. I find it appropriate to name them "Jurkat cells" or "Jurkat T cells" throughout the ms..

2. I am not convinced by the authors' assessment of the validity of the dSTORM and PALM data using AF647 and mEos (point Nr. 5 in my previous review). Fluorophore blinking and under-labeling are serious issues that in my view need to be accounted for. A simple yet compelling measure is to compare side by side the convolved (1) superresolution image with an ensemble image taken prior to bleaching / photo(re-)activation, as I have pointed out in my first report. Rossoth et al (Nat. Imm. 2018, e.g. Fig.1) from the Schütz lab have applied this approach to show how artifactual STORM and PALM localization maps can be. AF647 was part of the analysis. I can neither see why this is not relevant when attempting to co-localize to protein species, nor why this could not be done without great effort.

Authors: We would like to thank the reviewers again for their constructive evaluation of our manuscript and for their critical comments. We believe, that based on this, the changes and additions to the manuscript have improved its quality and intelligibility significantly. In this second revision, we have focused on the clarification of the two remaining critical issues: i) stronger evidence that glycine coating better prevents non-specific activation of Jurkat cells, and ii) evaluation of glycine-coating for primary T cells. Below, we respond to all the reviewer's comments in more detail.

Reviewer #1 (Remarks to the Author):

Thank you to the authors for answering most of the questions and for the manuscript changes that have increased its quality. We can divide the manuscript's content into three main topics, the use of glycine surface as a non-activatory surface, the development of the dTRABI method and the biological observation about CD4 and CD45 distribution at T cell protrusions on a resting state. I still have, however, a few significant concerns/comments regarding two of the topics that are as listed.

- Even if I understand the justification for my previous 1.3 and 1.4, I still believe that in order to confirm that the glycine surface is inert, this needs to be extensively characterised. There are many options to measure calcium release, including imaging-based approaches. Several groups have reported calcium release using confocal measurements, which can easily measure calcium dye fluorescence for hundreds of cells and record timelapse. This "activating" vs "inert" surface question is not critical for the establishment of dTRABI, however, all the biological conclusions made at the end of this manuscript are based on the assumption that glycine surfaces are inert. For example, in the new IRM data, the authors could label cells with calcium dye and follow calcium release as cells interact with both surfaces. This experiment would strengthen the IRM data and confirm that glycine is not simply slowing down contacts, for example (imaging data variability?). To make a clear case for the inert nature of the glycine surface, the authors should try to quantify calcium release over time, especially in the first 10-20 minutes (to fit with the immobilisation and fixation of cells described for SR imaging).

Authors: We agree with the reviewer that several approaches were reported to measure calcium mobilisation in cells using microscopy. The reason why we originally used genetically encoded GCaMP6f sensor is the fast kinetics with which this sensor interacts with calcium. That's why we still prefer to highlight these data in the main text. However, we agree that the number of cells we evaluated is not sufficient for strong conclusions. Therefore, we have monitored large numbers of cells using methods previously reported in the literature. Our efforts to acquire simultaneously IRM images and calcium-induced fluorescence were halted by the fact that the two techniques require different focal planes, which are not available in our current instrument setups. However, we have successfully applied low-resolution wide-field imaging of Fluo-4 in landing cells and analysed the data with the previously published algorithm for quantitative analyses of calcium response by microscopy, *i.e.* CalQuo2 (ref. 90 in the current manuscript).

Our new data support the original observation that glycine causes an important delay in non-specific stimulation of cultured Jurkat cells and that these cells respond less actively to the glycine-coated glass surface than any other treatment (new Supplementary Fig. S8). We have now analysed thousands of cells using wide-field fluorescence microscopy. Additionally, we followed isolated primary human CD4+ T cells during their immobilisation on glass surfaces coated with antibodies, PLL and glycine. Primary T cells responded vividly to anti-TCR antibodies, but much less in case of the PLL and glycine-coated surfaces. Moreover, our detailed analyses uncovered that

almost half of the primary T cells on glycine have not been activated at all. This result differs significantly from T cells that landed on PLL-coated coverslips, which were almost always activated (even though not as intensely as Jurkat cells; Supplementary Fig. S8). Again, several thousands of primary T cells were analysed using this approach. We believe that our data now strongly indicate that glycine coating helps to prevent non-specific T-cell activation during their landing on imaging glass.

Of note, we consistently and deliberately state in the manuscript that glycine coating reduces the impact of cell immobilisation on coverslips. We do not declare glycine-coated coverslips as an 'inert surface'. Our primary aim was to prevent (or reduce) the collapse of cell surface morphology.

- In Fig. 5, authors claim "strong tendency of wild-type CD4 to accumulate at one specific topographical level". Based on 5f, CD45 colocalises nicely with CD4 in these protrusions. The authors state "broader axial distribution of CD45 receptors than CD4 WT" but not lack/exclusion of CD45 at the tips. This observation seems to be in contrast with the claims/conclusions from the data shown in Fig.7. In Fig. 7, both proteins should be imaged with FP or antibody (as in Fig. 5). Keep in mind that CD4 is shorter than CD45. On top of that, CD4 is labelled with FP and CD45 with antibody, which could exacerbate minor differences in distribution. Differences in labelling approaches could justify why the data shown in Fig. 5 and Fig. 7 is somewhat different, and this variability should be addressed more carefully.

Authors: We apologise for any potential ambiguousness, but the data presented in Fig. 5f cannot be interpreted as "colocalization". The graph documents the axial distribution width of CD4 (and of its variant) and CD45. We could not provide 3D colocalization data here due to a current lack of a 2-channel dTRABI method, which we were previously working on, but that was halted due to the corona pandemic. Therefore, we agree that Figures 5 and 7 cannot be directly compared. However, CD4 distribution patterns acquired using antibody labelling (Fig.5) and a genetic approach (Fig. 7) are rather similar. Moreover, CD45 consistently exhibited similar 'exclusion zones' in all experiments (Figs. 4-7), which included 3 different super-resolution instruments and methods (3D SMLM - Wuerzburg, 2D SMLM - Dresden and SOFI - Prague). Accumulation of CD4 to the tips of microvilli and exclusion of CD45 from these zones was directly observed only in a few cells showing large protrusions as indicated in Fig. 7e (see annotated part of Fig. 7 below; red arrows). The text of the manuscript thus tentatively states that "we suspect that CD4 preferentially accumulates at the tips of membrane protrusions (e.g., microvilli) and CD45 to the shaft of these structures and their base at the plasma membrane". The data in Fig. 7 actually show that CD4 and CD45 colocalise in several areas (usually the shaft of microvilli) on the surface of resting Jurkat cells (see below; yellow arrows). As mentioned in the manuscript, our data here support the recently published observation that CD45 is excluded from the tips of T-cell microvilli (ref. 76).

Fig. 7 (selection). Red arrow symbols indicate CD4/CD45 exclusion zones, yellow colocalization zones.

- It is essential to discuss the possibility of CD45 being sterically excluded from glass/glycine-cell interfaces. One way to exclude the role of the extracellular region would be to take advantage of CD45 mutants with truncated extracellular domains, which seem to penetrate surface-cell contacts (described in Chang et al.).

Authors: As mentioned above, our data support the currently reported exclusion of CD45 from the tips of microvilli (ref. 76). This work also focused on the mechanism, by which CD45 is excluded from the tips of microvilli. The authors (Jung et al.) demonstrate that it is the transmembrane domain of CD45, which prevents the presence of this receptor in the tips of microvilli. Using truncated mutants (as suggested by the reviewer), the authors exclude the impact of the extracellular domain on CD45 organization on T-cell microvilli.

The main aim of this manuscript is to provide a new and holistic approach for surface receptor nanotopography studies.

All in all, while the dTRABI approach seems very interesting and has the potential to become a useful method in the field, the data that supports the conclusions for the two biological claims – activating vs inert surfaces and CD45/CD4 distribution – could greatly benefit from relatively simple but important additional experiments.

Reviewer #2 (Remarks to the Author):

I thank the Authors for carefully revising the manuscript. In the letter they have answered most of the points I raised, but I miss their incorporation in the revised work, specifically the following:

Ad 2: One could try Quartz crystal microbalance with dissipation monitoring.

Authors: We thank the reviewer for suggesting this intriguing concept. From our understanding, QCM-D could indeed be used to study the viscoelastic properties of the glycine layer, and the adhesion properties of landing cells on this surface. Unfortunately, none of the authors has any substantial experience with this technique, nor an instant access to it. Therefore, we decided not to include these experiments to the current study.

Ad 3: While the cases with two overlapping components (e.g. in Fig S10c ROI#20) reflect in a low p-p and z_w values, the presentations should also highlight the significance of the second component (e.g. the relative fraction f_2 is tiny in Fig S10c ROI#6), which is a key piece of information for the interpretation of the underlying distributions. Probably the easiest and most illustrative way would be to plot localisation counts (as Fig 5b,d,i) averaged over all ROI (aligned and normalised, as Authors deem appropriate). Alternatively, average f_2 per bin could be plotted as the second axis alongside the parameter histograms, or used to weigh the individual data contributions to these histograms. Please also discuss this in the text to complement the informativeness of all parameters. The text should help readers make sense of the data and numerical results, as nicely exemplified in some places [... signifying a strong tendency of wild-type CD4 to accumulate at one specific topographical level"], but missing in others [e.g. "Furthermore, the axial analysis of localizations revealed a more pronounced mean value of D_FWHM for CD4 CS1 (88 nm) than for CD4 WT (57 nm)"].

Authors: As suggested by the reviewer, we generated localization count plots averaged over all ROIs for all cells per type and include this as Supplementary Fig. S14.

Supplementary Figure S14. Normalized localization count plots averaged over all ROIs for all cells per type. a) All localizations of the manual selection of each cell (ROI#1 in each image, cf. Supplementary Fig. S11) were plotted as distribution along the z-axis, which was adjusted according to the mean value of the first Gaussian (Gaussian 1 in Fig. 5e). The aligned distributions of all ROIs per cell were subsequently summed up, leading to the global distribution for each tested receptor. b) Logarithmic plot of a). In total, 7.48×10^6 , 8.76×10^6 and 1.40×10^6 localizations were averaged for CD4 WT, CD4 CS1 and CD45, respectively

We revised the text in the manuscript accordingly: [...] The different axial distribution of the two CD4 variants on T cells was further confirmed by averaging the localizations for all cells tested (21 CD4 WT and 18 CD4 CS1 cells, Supplementary Fig. S14). More examples of three-dimensional

dTRABI images and their quantitative analysis for all three receptors are shown in Supplementary Fig. S15 and Supplementary Fig. S11, respectively.

Ad 5: After [“As a consequence, many localizations given by the localization software were rejected by TRABI.”] and similar statements, referring to the original method, the reader does not easily take home the message on dTRABI improvement on this issue. To me this is important and deserves better exposure in the text.

Authors: We agree with the reviewer and added further context to the original method (cf. Intensity based biplane imaging – dTRABI / The principle of dTRABI):

The quality of the resulting axial coordinate depended on the precision of the TRABI intensity measurement. In order to exclude any interference from neighboring emitters as well as temporal overlap with other fluorophores, a set of rigorous exclusion criteria for spots was employed. As the reference intensity I_R has to be determined with a large radius (towards 1 μm), an exclusion zone twice this radius had to be taken to search for any adjacent localization that would compromise the intensity estimation. As a consequence, many localizations given by the localization software were rejected by TRABI due to impaired estimation of I_R .

Ad 6: I fully appreciate the efforts of the Authors for rigorous quantitative image analysis, which should also result in more informed visual inspection. For meaningful interpretation and therefore better insights into biology, I think it would be beneficial to understand whether the differences between the samples come primarily from the overall cell-by-cell variations or from a small sub-set of cells. From the Authors' response I cannot see why ROIs can be quantitatively compared, but cells (sets of ROIs) cannot, so please discuss this in the text. Please also discuss the effects of different expression levels in the text.

Authors: Please see comment 3. The added Supplementary Fig. S14 underlines general differences between the mutants. As indicated, there is also an inherent cell-by-cell variation potentially linked to the transfection efficiency. In general, we aimed at collecting a sufficient number of cells in order to both showcase the inherent variations, but also the overall statistically relevant differences between sample and receptor types.

Ad 9: These numbers are definitely more informative than “nanometre resolution”, stated in the abstract – please better specify it there.

Authors: Although we agree with the reviewer, in that the total numbers are somewhat more illustrative, we believe that the abstract should describe the content of the article without too much detail. The phrase ‘nanometer resolution’ is commonly used in this context. Therefore, we have decided to keep the abstract unaltered. However, in the spirit of clarity, we specified the axial localisation precision of the original approach “(TRABI) method in combination with a biplane detection as a powerful three-dimensional imaging tool with nanometer precision (<20 nm)” as well as the new approach “Therefore, as demonstrated by Fourier Ring Correlation (FRC) of according data sets, we achieve an improved structural resolution while maintaining the same axial localization precision of better than 20 nm as with the original TRABI approach (Supplementary Figs. S9 and S10, see also Methods)” in the manuscript.

Reviewer #3 (Remarks to the Author):

In its current version the ms. addresses almost all issues I have raised previously. There are however 2 aspects that do not.

1. Jurkat T cells are not T cells. I find it appropriate to name them "Jurkat cells" or "Jurkat T cells" throughout the ms..

Authors: We agree with the reviewer. However, this work focuses on the distribution of receptors on membrane protrusions, which are present on both, primary T cells and Jurkat cells. The text of the manuscript was carefully revised to avoid confusion.

2. I am not convinced by the authors' assessment of the validity of the dSTORM and PALM data using AF647 and mEos (point Nr. 5 in my previous review). Fluorophore blinking and under-labeling are serious issues that in my view need to be accounted for. A simple yet compelling measure is to compare side by side the convolved (1) superresolution image with an ensemble image taken prior to bleaching / photo(re-)activation, as I have pointed out in my first report. Rossboth et al (Nat. Imm. 2018, e.g. Fig.1) from the Schütz lab have applied this approach to show how artifactual STORM and PALM localization maps can be. AF647 was part of the analysis. I can neither see why this is not relevant when attempting to co-localize to protein species, nor why this could not be done without great effort.

Authors: We appreciate the reviewer's thoroughness in this matter, but we respectfully disagree with this assessment. Alexa Fluor 647 and mEos are well studied and reliable photoswitches, which we used in optimal buffer and imaging conditions. There is no indication of under-labeling or photo-switching artefacts in our data. Although we do appreciate the work by Rossboth et al., we would like to indicate that a plethora of SMLM papers has been published with these specific dyes, none of which include the suggested comparison. We agree, that bleaching, photo-physical kinetics etc. should be checked for novel fluorophores where the photoswitching performance is unknown or hard to estimate, or if imaging techniques are used that are prone to photobleaching such as STED. In our case, however, we do not think it to be reasonable nor necessary to repeat all reported SMLM experiments in order to record diffraction limited fluorescence images prior to SMLM imaging – which we haven't done for the reported data. Nevertheless, in the spirit of cooperativity and goodwill, we performed basic comparisons of diffraction limited fluorescence images and the according SMLM data set (selection below). As it is clearly visible, there are no indications of similar artefacts as reported in the recommended work by Rossboth et al.

Finally, we would like to thank the reviewer for bringing this potential issue to our attention. We will add the standard of taking a prior fluorescence image of the structure of interest to our SMLM workflows in the future.

Comparison of SMLM images (left column) and their according diffraction limited fluorescence counterpart (right images). Depicted is the signal of CD4 WT and its CD4 CS variant, labelled by immunofluorescence as described in the main manuscript, in exemplary Jurkat cells after landing on the surface. There are no indications of artefacts as described in Rossboth et al.

Reviewers' comments:

Reviewer #1 (Remarks to the Author):

Thank you for addressing my comments/reviews and for the extensive justification on the rebuttal letter. The reinforced focus on the development of an approach for surface receptor nanotopography studies improved the manuscript significantly. This might be an important tool for other scientists in the field.

Reviewer #2 (Remarks to the Author):

The Authors have answered my comments, and I can therefore endorse the publication of this manuscript.

Reviewer #3 (Remarks to the Author):

I appreciate the changes made with regard to my first point concerning the terminology of T cells and Jurkat cells.

My second point has only been in part addressed. I understand the figures included in the rebuttal letter as a comparison between the STORM image and a diffraction limited image directly acquired with the camera (or is this a mathematically convolved rendering of the STORM image? It looks like it). As I have stated in my previous review, I find the comparison between the (i) STORM image, the (ii) Gauss-convolved rendering of the STORM image and (iii) the diffraction limited (unmodified raw) image taken by the camera (prior to starting the single molecule acquisitions to generate the STORM image) meaningful. In particular the comparison between (ii) and (iii) (with adjusted digital contrast) is telling but missing in the rebuttal.

The ms. will gain in credibility if these images are included in the ms. (e.g. as an extended figure).

Authors: We would like to thank the reviewers again for their constructive evaluation of our manuscript. We believe that by addressing this last comment, the manuscript will reach the standards for publication in *Communications Biology*.

Reviewer #3 (Remarks to the Author):

I appreciate the changes made with regard to my first point concerning the terminology of T cells and Jurkat cells.

Authors: Thank you.

My second point has only been in part addressed. I understand the figures included in the rebuttal letter as a comparison between the STORM image and a diffraction limited image directly acquired with the camera (or is this a mathematically convolved rendering of the STORM image? It looks like it). As I have stated in my previous review, I find the comparison between the (i) STORM image, the (ii) Gauss-convolved rendering of the STORM image and (iii) the diffraction limited (unmodified raw) image taken by the camera (prior to starting the single molecule acquisitions to generate the STORM image) meaningful. In particular the comparison between (ii) and (iii) (with adjusted digital contrast) is telling but missing in the rebuttal.

The ms. will gain in credibility if these images are included in the ms. (e.g. as an extended figure).

Authors: We have added Supplementary Fig. S18 to the manuscript. It compares TIRF images acquired prior to SMLM, with dSTORM images and Gauss-convolved dSTORM images of the same cells. Four examples are provided. Moreover, the Methods section was revised to note this addition (highlighted text at the page 17).